# Functional Consequences of Pathogenic Variants of the *GJB2* Gene (Cx26) Localized in Different Cx26 Domains

**DOI:** 10.3390/biom13101521

**Published:** 2023-10-13

**Authors:** Olga L. Posukh, Ekaterina A. Maslova, Valeriia Yu. Danilchenko, Marina V. Zytsar, Konstantin E. Orishchenko

**Affiliations:** 1Federal Research Center Institute of Cytology and Genetics, Siberian Branch of the Russian Academy of Sciences, 630090 Novosibirsk, Russia; maslova@bionet.nsc.ru (E.A.M.); danilchenko_valeri@mail.ru (V.Y.D.); zytzar@bionet.nsc.ru (M.V.Z.); keor@bionet.nsc.ru (K.E.O.); 2Novosibirsk State University, 630090 Novosibirsk, Russia

**Keywords:** hereditary hearing loss, *GJB2*, Cx26 domains, non-synonymous variants

## Abstract

One of the most common forms of genetic deafness has been predominantly associated with pathogenic variants in the *GJB2* gene, encoding transmembrane protein connexin 26 (Cx26). The Cx26 molecule consists of an N-terminal domain (NT), four transmembrane domains (TM1–TM4), two extracellular loops (EL1 and EL2), a cytoplasmic loop, and a C-terminus (CT). Pathogenic variants in the *GJB2* gene, resulting in amino acid substitutions scattered across the Cx26 domains, lead to a variety of clinical outcomes, including the most common non-syndromic autosomal recessive deafness (DFNB1A), autosomal dominant deafness (DFNA3A), as well as syndromic forms combining hearing loss and skin disorders. However, for rare and poorly documented variants, information on the mode of inheritance is often lacking. Numerous in vitro studies have been conducted to elucidate the functional consequences of pathogenic *GJB2* variants leading to amino acid substitutions in different domains of Cx26 protein. In this work, we summarized all available data on a mode of inheritance of pathogenic *GJB2* variants leading to amino acid substitutions and reviewed published information on their functional effects, with an emphasis on their localization in certain Cx26 domains.

## 1. Introduction

One of the most common forms of genetic deafness has been predominantly associated with pathogenic variants in the *GJB2* gene (gap junction protein, beta-2, 13q12.11, OMIM #121011), encoding transmembrane protein connexin 26 (Cx26). Different alterations of the *GJB2* sequence can produce a variety of clinical outcomes, including the most common non-syndromic autosomal recessive deafness, type 1A (DFNB1A, OMIM #220290), autosomal dominant deafness, type 3A (DFNA3A, OMIM #601544), as well as syndromic forms combining hearing loss and skin disorders (OMIM #149200, #602540, #148210, #148350, and #124500).

Connexin 26, comprising 226 amino acids, belongs to the human connexin protein family, containing 21 members classified according to their molecular weight, ranging from 23 to 62 kDa in size (Cx23–Cx62). Based on their sequence homology, the connexins can be divided into five subgroups (α, β, γ, δ, or ε) [1]. The connexins function as gap junction channels, which are critical for intercellular communication in the tissues of all multicellular organisms, by allowing direct exchange of ions and small (<1–1.5 kDa) molecules (diverse metabolites, second messengers, short RNAs) [1,2,3,4,5]. All connexins have a similar topology, with four α-helical transmembrane domains, two extracellular loops, a cytoplasmic loop, the N-terminus, and the C-terminus. A hexameric annular assembly (termed connexon), consisting of six connexin molecules, forms a pore through the lipid bilayer of the plasma membrane [6,7,8,9]. Different connexins are ubiquitously expressed in tissue- or cell-type-specific and overlapping patterns [10]. Connexons can be composed by connexins of either the same (homomeric connexons) or different types (heteromeric connexons). The ability of different types of connexins to oligomerize with each other seems to be determined by specific sequences in certain protein domains [11,12]. A connexon of one cell can contact (in a head-to-head disposition) another connexon of a neighboring cell, forming an intercellular channel. The aggregates of hundreds to thousands of gap junction channels form so-called gap junctional plaques on the plasma membrane. However, undocked or unopposed connexons can remain in the form of so-called hemichannels, which allow communication between the cellular interior and the extracellular space under certain physiologic and pathologic conditions [13,14,15,16,17,18].

The intercellular communication by different connexin gap junction channels and hemichannels regulates (by their specific permeability to metabolites) the signaling and function of various organs: endogenous expression of connexin family members can be mapped to over 110 distinct cell types found within all 12 human body systems [19]. Thus, the connexins play an important role in the development, physiology, and the etiology of several diseases. About half of the human connexins have currently been linked to a variety of genetic diseases caused by pathogenic variants in connexin-coding genes, and numerous reviews are devoted to different aspects of the connexins functioning in normal physiological and pathological processes [20,21,22,23,24,25,26,27,28,29,30,31,32,33].

## 2. Structure, Life Cycle, and Functions of Cx26

According to its sequence homology, Cx26 protein belongs to the β subgroup of connexins, which also includes Cx30 and Cx32. In 2009, Maeda et al. [8] first reported the structure of the Cx26 gap junction channel at high resolution (3.5 Å) obtained by X-ray three-dimensional crystallographic analysis. This model has greatly advanced the understanding of the overall architecture of connexin channels obtained by previous lower-resolution models [34,35,36] and became the basis for subsequent detailed biochemical and physiological analyses of connexin channels. The Cx26 domains were structurally resolved: four transmembrane helices (TM1–TM4), two extracellular loops (EL1 and EL2), a N-terminal region (NT), a cytoplasmic loop (CL), and a C-terminal tail (CT). Cx26 subunits assemble in hexamers and create an aqueous pore in the plasma membrane. The walls of pores are composed from TM1 and TM2 domains, and the NT regions of six subunits of Cx26 line the cytoplasmic pore entrance, forming a gradually narrowed funnel [8,37]. The mouth is created by CL and part of NT with several positively charged residues. Within the channel, TM1 and TM2 are covered by the first part of the NT, which is α-helical. The beginning of the NT is located deep within the pore. The NT helix extends beyond the cytoplasmic side of the membrane and then forms a loop that bends back to the membrane where TM1 begins. The bend appears to be flexible for the potentially rapid movements required for gating of the channel. The NT domain has been proposed (due its hydrophobic interactions with residues of the TM1 domain) to act as a gate-type “plug” whose movement is dependent on membrane voltage changes [8,35,38]. Each connexin protein has three intramolecular disulfide bonds between three pairs of cysteines in two extracellular loops, EL1 and EL2, which mediate the docking of hemichannels and are essential for the formation of the intercellular channel. On the extracellular side, Cx26 presents an accumulation of negatively charged residues [8,37]. The structure of the Cx26 channels was presented in more detail in further structural studies [9,39,40,41,42,43,44].

To date, there are numerous comprehensive reviews concerning different aspects (biosynthesis, post-translational modifications, assembly, trafficking, turnover, and channel function) of the life cycle of connexins [5,19,22,45,46,47,48,49,50,51]. The most complete information appears to come from the studies of connexin 43 (Cx43), which is natively expressed in nearly half of the more than 200 cell types found in the human body [19]. We aimed to briefly describe the life cycle of connexin 26, emphasizing its features distinctive from other connexins.

The expression of the *GJB2* gene, coding Cx26, is observed in many different tissues. In the human cochlea, *GJB2* is expressed by almost all cell types (except hair cells): supporting cells in the sensory epithelium, fibrocytes and mesenchymal cells in the lateral wall, basal and intermediate cells of the stria vascularis, and type I neurons in the spiral ganglion [52]. In the inner ear, besides Cx26, connexin 30 (Cx30) is predominantly expressed, and the expression of other connexins (Cx31, Cx43) was also detected [53,54,55,56]. Since Cx26 and Cx30 are both β-connexins, they are able to oligomerize and form heteromeric and heterotypic gap junction channels [57,58]. The β-connexins (including Cx26) immediately oligomerize in the endoplasmic reticulum, which differentiates them from α-connexins (e.g., Cx43), in which the oligomerization process is completed in the Golgi apparatus [59,60,61,62]. The α-connexins have an LR motif in TM3 and a WYIYGF motif in the EL2 domain, which allow them to interact with ERp29 and some other chaperones, which stabilize connexin molecules as monomers until they get to the Golgi apparatus. Since β-connexins have a WW motif in the TM3 domain and an FYxLYxG motif in the EL2 domain, they are extremely unstable and have no opportunity to bind ERp29, thereby immediately oligomerizing in the endoplasmic reticulum [11,12,59,60,61,62].

Fully oligomerized connexons transport to the plasma membrane of the cell. The transport pathways have not yet been fully elucidated. Some studies reported on existing both Golgi-dependent and microtubules-independent variants [63,64,65]. There are studies both confirming the participation of actin in the traffic of connexons formed by Cx26 [63,66] and refuting this [65,67].

The gap junction plaque is a dynamic structure due to the half-life of connexin 26 of about 1.5–5 h; therefore, the newly formed connexons substitute the old ones from the periphery to the center of the plaque. It is known that most of the connexin proteins are transported from the periphery to the center of gap junction plaques via the actin-binding proteins ZOs (zonula occludens), ubiquitous scaffolding proteins, which are necessary for the delivery and organization of target proteins into complexes on the cell membrane [68,69]. The short amino acid sequence on the C-terminus of target proteins is usually recognized by the PDZ domain of ZOs, and thus allows the interaction between them [69]. Unlike many other connexins, Cx26 appears to lack a PDZ-binding motif [68], so its organization into gap junction plaques may apparently be regulated by other proteins [70].

The post-translation modifications of connexins were comprehensively reviewed by Aasen et al. [47]. Unlike many other connexins, the C-terminal domain of Cx26 is not phosphorylated, though its N-terminus is subjected to post-translational modifications, including acetylation of Met1 and Lys15, hydroxylation of Asn14, and phosphorylation of multiple potential phosphorylation sites [47,71]. Regulation of the Cx26 protein level is critical for its function, but the molecular mechanisms involved are only partially understood. In addition, at least twelve different proteins are able to interact with the C-terminus of Cx26, but the outcome of each interaction remains to be seen [72].

## 3. Current Hypotheses on the Mechanisms of Cx26-Associated Hearing Impairment

Recently, Chen et al. [50] reviewed the current concepts of the mechanisms of Cx26-associated hearing impairments. A decade ago, researchers believed that the failure in K^+^-recycling was the crucial role in the pathogenesis of Cx26-associated hearing loss. Accumulation of K^+^ ions near hair cells in the inner ear caused by non-functional gap junctions was proposed to produce cytotoxicity and the hair cells’ ablation, and the interruption of K^+^-recycling leads to a reduction of endocochlear potential and impairment of active cochlear amplification. However, this hypothesis cannot explain the pathogenesis of Cx26-associated hearing loss due to Cx26 variants that do not lead to the formation of dysfunctional gap junction channels. Another recent hypothesis proposes that Cx26-caused hearing loss could be associated with impairment of ATP-triggered intercellular Ca^2+^ signaling transduction, which plays an important role in postnatal auditory development. The third hypothesis proposes that the mechanism of Cx26-related hearing loss could be associated with nutrient function. Hemichannels and gap junction channels allow the passage of many components with molecular weight less than 1.2 kDa, including glucose, thereby providing necessary nutrients and energy for developing supporting cells of the inner ear. The lack of glucose leads to a reduction in ATP production, followed by excess accumulation of reactive oxygen species and cell apoptosis. None of these proposed hypotheses can yet fully explain the subtle mechanisms of the pathogenesis of Cx26-related hearing impairment, since hemichannels and gap junction channels formed by the Cx26 protein appear to play multiple roles in the development and function of inner ear structures [50].

## 4. Pathogenic Variations in the *GJB2* Gene Sequence

The *GJB2* gene encoding Cx26 (226 a.a.) contains two exons and one intron, with a coding region (681 nucleotides) in exon 2. Different alterations of the *GJB2* sequence can produce a variety of clinical outcomes, including the most common non-syndromic autosomal recessive deafness, 1A type (DFNB1A, OMIM #220290), autosomal dominant deafness, type 3A (DFNA3A, OMIM #601544), as well as syndromic forms, in which hearing loss is accompanied by mild to severe skin disorders: palmoplantar keratoderma with deafness (OMIM #148350), keratitis-ichthyosis-deafness syndrome (KID, OMIM #148210), Vohwinkel syndrome (OMIM #124500), Bart–Pumphrey syndrome (OMIM #149200), and hystrix-like ichthyosis with deafness (OMIM #602540).

The current information on the influence of pathogenic *GJB2* variants on the structure and functions of the Cx26 protein has recently been updated in several comprehensive reviews [50,73,74,75,76,77]. These data, together with bioinformatics and structural modeling studies [40,78,79,80,81,82], provide a basis for understanding the details of the molecular mechanisms by which pathogenic *GJB2* variants lead to different clinical hearing loss phenotypes.

We analyzed the variability of the *GJB2* gene sequence based on the data from the Deafness Variation Database (the DVD v9 version: https://deafnessvariationdatabase.org/, accessed on 1 June 2023) [83], which includes all known genetic variants presented in 223 deafness-associated (non-syndromic and syndromic) genes. The DVD integrates all available genetic, genomic, and clinical data together with expert curation to generate a single classification for each variant in the genes implicated in deafness [83]. For categorization of variants according to the standards and guidelines for the interpretation of sequence variants [84] (P, pathogenic; LP, likely pathogenic; B, benign; LB, likely benign; VUS, variants of unknown significance), the DVD uses available information from ClinVar and/or the published literature on PubMed, an assessment of functional significance and conservation of missense variants by several computational methods, and the data on MAF (minor allele frequency). Within the different classification categories (P, LP, VUS, LB, and B), the DVD also provides a molecular profile of the variants (UTRs, nonsense, missense, intronic, synonymous, splice-site, frameshift indels, start/stop loss, and in-frame indels).

To date, in total, 1165 different variants (P, LP, VUS, LB, and B) in the *GJB2* gene sequence have been reported on the DVD (https://deafnessvariationdatabase.org/gene/GJB2, accessed on 20 June 2023). Among them, variants classified as pathogenic or likely pathogenic (PLP variants) (*n* = 428) make up a significant proportion (total of 36.8%) (Figure 1). The PLP variants include different point mutations, small insertions and deletions, as well as several large deletions that remove the whole *GJB2* gene or regulatory sequences that are needed for the expression of *GJB2*. Among the PLP variants, the common molecular alterations are the missense variants (65.7%), followed by the frameshift variants (18.9%) (Figure 1).

Here, we focused on the PLP missense variants in the *GJB2* coding region leading to amino acid substitutions (*n* = 281), which have been found in all domains of Cx26 (Figure 2). It should be noted that many amino acids can undergo more than one substitution. The distribution of these variants across Cx26 domains appears to be uneven. In order to define Cx26 domains with the highest load of the PLP missense variants, the number of them in each Cx26 domain was normalized by the number of amino acids in the corresponding domain. Four Cx26 domains (N-terminus, TM1, EL1, and TM2) were found to have a rate above the mean (median value = 1.09) (Figure 3).

Numerous in vitro studies were performed by various experimental assays to elucidate the functional consequences of the PLP variants leading to amino acid substitutions; however, not all detected variants have been subjected to such studies, and the mechanism of their effect on the structure and function of Cx26 remains unclear. In the following sections, we review the PLP missense variants that have been analyzed in in vitro studies, in order of their location in the Cx26 protein domains.

We also analyzed and summarized all available literature data on the types of inheritance of PLP missense variants (Figure 4).

Among them, there are variants with evident autosomal recessive inheritance and autosomal dominant variants, which include non-syndromic (isolated hearing impairment) and syndromic variants (hearing impairment is accompanied by mild to severe skin disorders). However, for rare (found only in single patients) and poorly documented variants, information about the type of inheritance is not yet available (Figure 4).

### 4.1. N-Terminus (NT)

The N-terminus (NT) of Cx26 begins with the methionine encoded by the ATG translational start codon. This region includes 20 amino acids, consisting of intramembrane (from 1 to 13 a.a.) and cytoplasmic (from 14 to 20 a.a.) parts (Figure 2). According to the DVD, in NT, there are 27 different PLP missense variants (13 and 14 in intramembrane and cytoplasmic parts, respectively) leading to amino acid replacements. It should be noted that five amino acids can undergo more than one substitution: Gly12 (4), Asn14 (3), Ser17 (2), Ser19 (3), and Ile20 (3). There are non-syndromic PLP variants with an obvious autosomal recessive type of inheritance, quite a lot of variants with an uncertain type of inheritance, and five syndromic dominant variants (Figure 4).

#### 4.1.1. Recessive Variants

Five non-syndromic recessive PLP variants (p.Thr8Met, p.Leu10Pro, p.Gly12Val, p.Asn14Asp, and p.Ser19Thr) were analyzed in several functional studies [85,86,87,88,89,90].

Meşe et al. (2004, 2008) analyzed variant p.Thr8Met, along with other Cx26 mutations associated with non-syndromic recessive hearing impairment, in the paired *Xenopus* oocyte expression system. The Cx26-p.Thr8Met mutant formed partially functional channels with altered voltage-gating properties, suggesting that these channels may retain high permeability to potassium ions [85]. Further examination of the permeability of the p.Thr8Met mutant channels (along with other Cx26 mutants) revealed a differential selectivity of mutant channels for molecules with different molecular weights and charges [86].

Dalamon et al. performed electrophysiological and permeability studies of variant p.Leu10Pro [87]. The Cx26-Leu10Pro mutant protein can form functional channels and hemichannels, albeit with different voltage dependence and solute permeability properties (decreased ATP/cationic dye selectivity) than those formed by wtCx26. Molecular dynamics simulations revealed the changes in the pore structure and electrostatic charges in the Cx26-Leu10Pro mutant that could be involved in the decreased dye uptake and ATP release [87].

The recessive variants p.Gly12Val and p.Ser19Thr were analyzed in the study by d’Andrea et al. [88]. Both variants showed a slower electrophoretic mobility in comparison to wtCx26 and a lack of appreciable membrane staining. The product of the p.Gly12Val variant results in a complete intracellular retention of the protein, clustered in large perinuclear vesicles, which is accompanied by a significant decrease of protein expression. The mechanism affecting the functionality of the p.Ser19Thr mutant appears different, in that at least some Cx26 is properly routed to the cell surface [88]. García et al. analyzed the functional state of channels formed by different homomeric mutant Cx26s and revealed that p.Gly12Val completely abolished the formation of functional channels [89].

Variant p.Asn14Asp was detected in compound heterozygosity with pathogenic variant c.35delG in two brothers with moderate non-syndromic sensorineural hearing impairment [90]. Functional analysis of mutant Cx26 expressed in *Xenopus laevis* oocytes showed that the p.Asn14Asp mutant protein was synthesized properly but integrated only in markedly reduced amounts into the cell membrane. The Cx26-Asn14Asp expressing oocytes showed a significantly reduced current compared to wild-type Cx26-injected cells. In addition, the co-injection of wild-type and mutant cRNA at equimolar levels restored the conductive properties, supporting the recessive character of p.Asn14Asp [90].

#### 4.1.2. Dominant Variants

All dominant variants in NT (p.Gly11Glu, p.Gly12Arg, p.Asn14Tyr, and p.Ser17Phe) are syndromic and associated with keratitis-ichthyosis-deafness (KID) syndrome (OMIM #148210) or with clinical phenotypes overlapping with KID syndrome (p.Asn14Lys).

**Gly11.** Variant p.Gly11Glu, first found in a patient with KID syndrome, was analyzed in the studies by Terrinoni et al. [91,92]. The expression of the Cx26-Gly11Glu mutant protein was significantly reduced when compared to normal Cx26. The deficiency in functional connexon formation by this mutant protein results in aberrant calcium gating, leading to increased intracellular calcium and necrotic cell death [91,92].

**Gly12.** All beta-connexins include a glycine residue at position 12 (Gly12), and mutations that alter Gly12 in different connexins are associated with skin disease as well as hereditary forms of deafness and neuropathy [93]. In Cx26, four substitutions at Gly12 are known. One of them, p.Gly12Arg, is associated with KID syndrome, while three other variants (p.Gly12Val, p.Gly12Asp, and p.Gly12Cys)—with non-syndromic hearing loss.

Syndromic variant p.Gly12Arg was analyzed in several studies [89,94,95,96]. In the study by Lee et al. (2009), the effects of p.Gly12Arg have been explored in the *Xenopus* oocytes expression system. The Cx26-Gly12Arg mutant protein demonstrated significantly increased hemichannel activity compared to the wild-type protein correlated with an increased cell death. When cultured in solutions with higher extracellular Ca^2+^ concentrations, the p.Gly12Arg hemichannels had reduced levels of activity and increased cell survival [94]. García et al. (2015) investigated the effects of p.Gly12Arg, along with other Cx26 pathogenic variants, when it was co-expressed with other Cxs (in particular, Cx43). They demonstrated that p.Gly12Arg changes the connexin oligomerization compatibility, allowing aberrant interactions with Cx43. A heteromeric oligomer formed by Cx43 and the p.Gly12Arg-mutant Cx26 showed exacerbated hemichannel activity but non-functional channels. Heterologous expression of these hyperactive heteromeric hemichannels increases cell membrane permeability, favoring ATP release and Ca^2+^ overload [89]. García et al. (2018) investigated the biophysical properties of the syndromic mutant Cx26-Gly12Arg expressed in *Xenopus laevis* oocytes and the structural changes induced by p.Gly12Arg by molecular dynamics simulations. The NT is an active component of both the slow and fast gating mechanisms. At the molecular level, the p.Gly12Arg appears to allow an interaction between Gly12 and Gly99, fixing NT in a new position toward the cytoplasm, thereby preventing its action as a blocking gating particle. Disruption of this interaction recovers the fast and slow voltage-dependent gating mechanisms [95]. Albuloushi et al. (2020) analyzed, by using a range of cell-based assays, the functional effects of p.Gly12Arg (along with p.Phe142Leu and p.Asp66His) in HeLa cells expressing Cx26 or Cx43 and in HaCaT cells, a model keratinocyte cell line. In HeLa26 cells co-expressing p.Gly12Arg, the mutant protein was restricted to perinuclear areas. The viability of the p.Gly12Arg-transfected cells was greater in HeLa cells expressing Cx43 than in native Cx-free HeLa cells. Expression of p.Gly12Arg results in microtubule collapse, rescued by interaction with Cx43 [96].

**Asn14.** The amino acid asparagine (Asn, N) at position 14 is highly conserved among different species and different types of connexins. There are three different amino acid substitutions at Asn14: p.Asn14Tyr and p.Asn14Lys, leading to syndromic (hearing loss and skin anomalies), and p.Asn14Asp leading to non-syndromic hearing loss. Several studies were performed to elucidate the functional consequences of these variants by using different experimental assays [89,94,97,98,99,100,101].

Arita et al. (2006) analyzed variant p.Asn14Tyr found in a patient with KID syndrome, in the cell culture obtained by biopsy from a hyperkeratotic plaque in the patient’s skin. By using experimental assays and molecular structural analysis (NMR), they concluded that p.Asn14Tyr appears to induce a change in the local structural flexibility of the N-terminal domain, which is important for exerting the activity of the channel function, resulting in impaired gap junctional intercellular communication [97]. Later, the effects of p.Asn14Tyr were studied by Lee et al. [94]. The mutant protein was expressed in *Xenopus* oocytes with levels equal to wild-type Cx26. The p.Asn14Tyr resulted in larger hemichannel currents than the wild-type-expressing cells, and elevated hemichannel activity correlated with an increased cell death. The cell death phenotype of p.Asn14Tyr could be rescued through the addition of extracellular Ca^2+^. The voltage-gating sensitivity of junctions formed by p.Asn14Tyr channels was greatly reduced compared to wild-type Cx26 [94].

De Zwart-Storm et al. (2011) assessed, by immunofluorescence staining and the parachute assay, the functional consequences of p.Asn14Tyr and p.Asn14Lys and revealed that these variants have different protein localization and gap junction permeability. The p.Asn14Lys causes a pronounced cytoplasmic accumulation that can be partially rescued by co-transfection with wild-type Cx26, and cells containing the mutant p.Asn14Lys are unable to transfer the dye to recipient cells. In contrast, the p.Asn14Tyr is readily transported and inserted into the cell membrane, and cells expressing mutant p.Asn14Tyr protein were able to transfer the dye [98]. García et al. (2015) showed that p.Asn14Tyr co-localized with Cx43 in intracellular compartments and channel plaques, suggesting aberrant formation of heteromeric channels (Cx43/Cx26 mutants). In addition, p.Asn14Tyr produced heteromeric hemichannels with increased activity, enhancing the cell membrane permeability [89]. In electrophysiological experiments using *Xenopus* oocytes, Sanchez et al. (2016) showed that p.Asn14Lys and p.Asn14Tyr, when expressed alone or together with wild-type Cx26, result in functional hemichannels, which exhibit substantial differences in functional efficiency, gating, and sensitivity to pH [99].

Press et al. (2017) investigated p.Asn14Lys, along with other syndromic or non-syndromic dominant Cx26 variants, and demonstrated that when gap-junction-deficient HeLa cells expressed the p.Asn14Lys, they underwent cell death, such that meaningful channel function information was unattainable [100]. Valdez Capuccino et al. (2019) explored (by combining electrophysiological techniques, double-mutant cycle analysis, and MD simulations) the molecular basis by which p.Asn14Lys promotes gain of function. They found that the p.Asn14Lys mutant favors the open conformation of hemichannels, shifts calcium and voltage sensitivity, and slows deactivation kinetics. The MD simulations of WT and p.Asn14Lys hemichannels indicated that the p.Asn14Lys variant increases the hemichannel open probability by disrupting interactions between the NT and the TM2/CL region of the adjacent connexin subunit [101].

**Ser17.** Alterations in the Cx26 amino acid sequence at position 17 (Ser17) can lead to KID syndrome (p.Ser17Phe) or non-syndromic hearing loss (p.Ser17Tyr) [102]. Thus, despite that in both cases serine is replaced by a more bulky aromatic amino acid such as phenylalanine (p.Ser17Phe) or tyrosine (p.Ser17Tyr), the consequences can be significantly different. Richard et al. (2002) revealed that coupling-incompetent HeLa cells expressing the p.Ser17Phe mutant Cx26 were completely uncoupled, unlike the cells with wild-type Cx26, indicating that p.Ser17Phe abolishes the function of mutant Cx26 in vitro [103]. Mazereeuw-Hautier et al. (2014) reported a case of lethal KID syndrome associated with p.Ser17Phe. In this family, p.Ser17Phe probably appeared de novo since the parents of the patient were not carriers of pathogenic *GJB2* variants [104]. García et al. (2015) demonstrated that p.Ser17Phe, as well as other syndromic mutations (p.Gly12Arg and p.Asn14Tyr) in NT of Cx26, changes the connexin oligomerization compatibility by aberrant interactions with Cx43 [89]. Abbott et al. (2023) found that cochlear tissue expressing Cx26-Ser17Phe presented increased membrane permeability in the supporting cells of the organ of Corti through the formation of hyperactive hemichannels, producing damage of hair cells in the organ of Corti. The mutant Cx26-Ser17Phe can associate with Cx30, producing hyperactive hemichannels with reduced sensitivity to extracellular Ca^2+^ and standard hemichannel blockers. Molecular dynamic models of heteromeric mutants show alteration in the residues involved in Ca^2+^ coordination [105].

### 4.2. Transmembrane Domain 1 (TM1)

Transmembrane domain 1 (TM1) of Cx26 includes 20 amino acids (from 21 to 40 a.a.). There are 32 PLP variants. Among them, a little over 40% are non-syndromic variants with obvious autosomal recessive inheritance, three variants are dominantly inherited syndromic variants, associated with KID syndrome or deafness with palmoplantar keratoderma (PPK) (OMIM #148350), and the remaining variants have an unclear type of inheritance (Figure 4). Multiple substitutions were found at six amino acid positions of the TM1 sequence: Ile30 (3), Arg32 (5), Ile33 (2), Met34 (5), Val37 (4), and 40Ala (4).

**Ile30**. There are three different amino acid substitutions at Ile30: p.Ile30Asn, leading to KID syndrome, and p.Ile30Val and p.Ile30Leu, leading to non-syndromic hearing loss, which have not been functionally studied. Aypek et al. (2016) analyzed the effect of syndromic variant p.Ile30Asn on the protein biosynthetic pathway, hemichannel activity, and intracellular calcium levels in communication-deficient HeLa cell lines [106]. The p.Ile30Asn resulted in the retention of mutant protein in the Golgi apparatus and failed to form gap junction plaques at cell–cell contact sites. Cells with the Cx26-Ile30Asn variant had increased dye uptake compared to wtCx26-containing cells, indicating abnormal hemichannel activities. Cells with mutant protein had elevated intracellular calcium levels compared to wtCx26-transfected cells [106].

**Arg32, Ile33, Ile35.** Non-syndromic autosomal recessive variants p.Arg32His, p.Ile33Thr, and p.Ile35Ser were analyzed by several experimental assays to observe the mutant proteins’ localization, channel formation ability, and channel permeability to fluorescent dyes in transiently transfected HeLa cells [107,108] or in HEI-OC1 cells derived from the progenitor epithelium of the organ of Corti from mice [109]. Mani et al. (2009) showed that the p.Ile35Ser mutant protein exhibits impaired trafficking (found largely in the cytoplasm), whereas the p.Ile33Thr mutant exhibited membrane localization similar to wild-type Cx26, although intercellular transfer of LY (Lucifer yellow) between p.Ile33Thr cells was not observed [107]. The p.Arg32His variant failed to reach the cell surface and co-localized with the endoplasmic reticulum throughout the cells [108]. Beach et al. (2020) revealed that p.Arg32His did not form clearly identifiable gap junction plaques but appeared to be partially localized within intracellular vesicles [109].

**Met34, Val37.** Two variants, p.Met34Thr (c.101T>C) and p.Val37Ile (c.109G>A), are notable missense *GJB2* variants because their classification has long been controversial, despite that these variants have been found in a significant number of patients with hearing loss [110]. The p.Met34Thr was first reported as being associated with dominant hearing loss [111] and the p.Val37Ile was first identified as a polymorphism in a heterozygous control [112]. Subsequently, p.Met34Thr was found in individuals with normal hearing, which suggests an autosomal recessive mode of inheritance; in turn, p.Val37Ile was later found to be homozygous or *in trans* with known pathogenic *GJB2* variants in affected individuals. Both variants were found relatively frequently in the general population. The phenotypic manifestation of hearing loss related to p.Met34Thr or p.Val37Ile varied from apparently normal to profound. However, these two variants were typically associated with bilateral mild to moderate sensorineural hearing loss, affecting high- to mid-sound frequencies [110]. Reduced penetrance of these two variants has been proposed and later confirmed [110,113].

Numerous in vitro studies have investigated the functional effects of p.Met34Thr on gap junction function via analysis of its intracellular localization, oligomerization, and dye-loading assays. The p.Met34Thr mutant protein exhibited strong staining at the plasma membrane and minor evidence for intracellular retention [88,114,115]. Very low or no dye (LY) transfer was observed between cells expressing p.Met34Thr [88,115,116], while intercellular neurobiotin diffusion was present [117]. The oligomerization assays showed that the p.Met34Thr mutant failed to efficiently assemble into hexameric gap junction hemichannels [116,117]. Homotypic p.Met34Thr/p.Met34Thr channels did not induce intercellular coupling in *Xenopus* oocytes, and heterotypic pairings with Cx26 (Cx26/M34T) revealed functional channels with gating properties significantly different than those of wtCx26 [114]. The p.Met34Thr inefficiently formed intercellular channels that displayed an abnormal electrical behavior and the Met34Thr channels failed to support the spreading of mechanically induced intercellular Ca^2+^ waves [115]. Co-expression of mutant Cx26-Met34Thr significantly decreased the wild-type Cx26-mediated currents when wild-type and mutated Cx26 cRNAs were co-injected in *Xenopus* oocytes at equimolar levels, resembling the situation in heterozygotic individuals [118]. Oshima et al. (2003) proposed an M34-regulating model for the fully open state of the gap junction channel, in which residue Met34 makes important interactions with other transmembrane helices [119]. Using molecular dynamics simulations, Zonta et al. (2014) indicated that the quaternary structure of the Cx26-Met34Thr hemichannel is altered at the level of the pore funnel due to the disruption of the hydrophobic interaction between M34 and tryptophan3 (Thr 3) in the N-terminal helix [120].

Variant p.Val37Ile did not induce the formation of homotypic intercellular channels in the paired *Xenopus* oocyte expression system [121]. Electrophysiological analysis of the channel activity revealed reduced conductance when p.Val37Ile was co-expressed with wild-type Cx26, Cx30, or Cx31 [118]. The p.Val37Ile did not show alteration in the intracellular protein expression and was localized at the cell membrane, where it formed gap junctions between two adjacent cells; however, biochemical coupling was significantly reduced in Cx26-Val37Ile-transfected cells. In the hemichannel study, the number of dye-loaded cells was greatly reduced in Cx26-Val37Ile-transfected cells [122].

Thus, based on compelling statistical and supporting functional evidence, p.Met34Thr and p.Val37Ile were classified as pathogenic autosomal recessive variants with incomplete penetrance [110].

**Ala40.** There are four different substitutions at position 40Ala of Cx26, which can lead to non-syndromic recessive hearing loss (p.Ala40Glu and p.Ala40Gly), hearing loss with an uncertain mode of inheritance (p.Ala40Ser), or to KID syndrome (p.Ala40Val). The p.Ala40Val variant was first discovered in a patient with KID syndrome, in whom it appears to have arisen de novo [123]. Expression of the Cx26-Ala40Val in *Xenopus* oocytes resulted in a disorganization of cell pigmentation, followed by oocyte death, which correlated with a markedly decreased plasma membrane electrical resistance [123]. Zhang et al. (2005) analyzed, by two-electrode patch-clamp recordings, the effect of p.Ala40Glu on the properties of gap junction channels and revealed that p.Ala40Glu results in completely disrupted intercellular electrical coupling [124]. In a subsequent study in the *Xenopus* oocyte expression system, the ability of elevated Ca^2+^ to rescue p.Ala40Val was tested. It was shown that the addition of Ca^2+^ to the media produced a dramatic reduction in the Ala40Val hemichannels’ current flow [125]. Sánchez et al. (2010) quantified the Ca^2+^ sensitivities and examined the biophysical properties of the p.Ala40Val mutant at macroscopic and single-channel levels. The p.Ala40Val hemichannels showed significantly impaired regulation by extracellular Ca^2+^, which is likely the principal mechanism that contributes to disease in the case of p.Ala40Val [126].

In the study by Jara et al. (2012), individual transmembrane (TM) domains and the full-length Cx26 protein were studied to identify motifs involved in oligomerization of Cx26. Using different experimental assays, they showed that TM1 had a strong propensity to homodimerize and the TM1 amino acids motif Val-37–Ala-40 (VVAA) is required for homodimerization. Two deafness-associated Cx26 mutations localized in this region, p.Val37Ile and p.Ala40Gly, differentially affected dimerization. TM1-Val37Ile dimerized only weakly, whereas TM1-Ala40Gly did not dimerize. When the full-length mutants were expressed in HeLa cells, both Cx26-Val37Ile and Cx26 Ala40Gly formed oligomers less efficiently than wild-type Cx26. Unlike wild-type Cx26, neither Cx26-Val37Ile nor Cx26-Ala40Gly formed functional hemichannels in low extracellular calcium. Thus, the VVAA motif of Cx26 is critical for TM1 dimerization, hexamer formation, and channel function. The differential effects of VVAA mutants on hemichannels and gap junction channels imply that inter-TM interactions can differ in unopposed and docked hemichannels. Moreover, Cx26 oligomerization appears dependent on transient TM1 dimerization as an intermediate step [62].

### 4.3. Extracellular Loop 1 (EL1)

Among all Cx26 domains, the maximum number (*n* = 64) of different PLP variants resulting in amino acid substitutions was found in the extracellular loop 1 (EL1), including 33 amino acids (from 41 to 73 a.a.). There are 18 amino acids that undergo more than one replacement. This Cx26 domain is also distinguished by a maximum number of dominant variants (*n* = 23), both non-syndromic (*n* = 11) and syndromic (*n* = 12) (Figure 4). All syndromic PLP missense variants can be associated with deafness and skin pathologies, often with overlapping clinical phenotypes and varying penetrance: deafness with PPK (OMIM #148350), KID syndrome (OMIM #148210), overlapping with hystrix-like ichthyosis with deafness (OMIM #602540), Vohwinkel syndrome (OMIM #124500), and Bart–Pumphrey syndrome (OMIM #149200).

#### 4.3.1. Recessive Variants

Functional in vitro studies were performed for almost all known dominant (non-syndromic and syndromic) variants in the EL1 domain, while only four recessive variants (p.Glu47Lys, p.Glu47Gln, p.Thr55Asn, and p.Ile71Asn) were subjected to such studies.

Stong et al. (2006) investigated the properties of two variants, recessive p.Glu47Lys and syndromic dominant p.Gly45Glu, expressed in HEK293 cells, and analyzed the ionic and biochemical permeability of reconstituted gap junctions and hemichannels. Both variants affected the gap junction functions in dramatically different manners. The Cx26-Glu47Lys mutant protein formed a non-functional gap junction that lacked channel- and hemichannel-mediated biochemical and ionic coupling. In contrast, p.Gly45Glu resulted in apoptosis and cell death that was rescued by increasing the concentration of extracellular calcium in a dose-dependent manner [127]. Using ionic and biochemical coupling tests and computational analyses to predict structural abnormalities, Kim et al. (2016) evaluated the pathogenic effects of ten different Cx26 variants. For seven of them, a loss of gap junction function was predicted, whereas the others would completely fail to form gap junction channels. Functional studies demonstrated that while all variants were able to function normally as hetero-oligomeric channels, six variants, including p.Glu47Lys, did not form functional homo-oligomeric gap junction channels [81].

Melchionda et al. (2005) reported a family from Southern Italy with non-syndromic post-lingual hearing loss, due to a novel missense variant p.Thr55Asn. Functional studies indicated that p.Thr55Asn produces a protein that, although it demonstrated an expression level similar to wtCx26, its intracellular trafficking is deeply impaired, and it fails to reach the plasma membrane [128].

Mohamed et al. (2010) found a new missense variant, p.Ile71Asn, in a patient from Southern Egypt and tested its functional properties in *Xenopus laevis* oocytes, where the Cx26 hemichannel activity was measured by depolarization-activated conductance in non-coupled oocytes. As a result, the Ile71Asn-mutated channel turned out to be non-functional [129].

#### 4.3.2. Dominant Variants

Four dominant non-syndromic (hearing impairment without additional visible clinical traits) variants at positions 44Trp, Gly45, and Asp46 (p.Trp44Cys, p.Trp44Ser, p.Gly45Arg, and p.Asp46Glu) were analyzed in several functional studies [116,130,131,132,133,134,135,136].

Martin et al. (1999) investigated the functional effects of three mutant Cx26 variants in an in vitro expression system, including p.Trp44Cys, by examining key steps involved in the assembly of gap junctions. The p.Trp44Cys efficiently assembled into hexameric gap junction hemichannels and targeted to the plasma membrane, but no dye (LY) transfer was observed in cells expressing p.Trp44Cys; thus, p.Trp44Cys resulted in impaired intercellular coupling [116]. Bruzzone et al. (2001) characterized the ability of p.Trp44Cys to form channels in paired *Xenopus* oocytes and transfected HeLa cells. Oocyte pairs injected with p.Trp44Cys were not electrically coupled above background levels, and p.Trp44Cys failed to dye couple-transfected HeLa cells. Moreover, p.Trp44Cys dramatically inhibited intercellular conductance of wtCx26 when co-expressed in an equal ratio, and the low levels of residual conductance displayed altered gating properties [130]. Rouan et al. (2001) studied the effects of variant p.Trp44Cys in paired *Xenopus* oocytes, along with other Cx26 mutants identified in individuals with hearing loss and palmoplantar keratoderma, on gap junctional intercellular communication. All analyzed Cx26 mutants were functionally impaired and failed to induce intercellular coupling. When co-expressed with wtCx26, all four mutants suppressed the wtCx26 channel activity, consistent with a dominant inhibitory effect [131]. Marziano et al. (2003) investigated the properties of four Cx26 mutants derived from several variants associated with dominantly inherited hearing loss, either non-syndromic (p.Trp44Ser, p.Arg75Trp) or with various skin disorders (p.Gly59Ala, p.Asp66His). They also studied the effects of these mutant variants on wild-type Cx26 and Cx30 proteins, which are co-localized within the inner ear. The p.Trp44Ser mutant, observed by immunohistochemistry, was localized to the plasma membrane. Dye transfer studies demonstrated a disruption of the intercellular coupling for all four of the mutant proteins. These results indicate the effect of p.Trp44Ser on the ability to form properly functional channels. In addition, the p.Trp44Ser mutant demonstrated a significantly reduced dye transfer rate between cells co-expressing either Cx26 or Cx30 together with p.Trp44Ser compared with the wild-type proteins alone [132]. The interactions of nine dominant Cx26 mutants, six associated with non-syndromic hearing loss, including p.Trp44Cys and p.Trp44Ser, and three associated with hearing loss and palmoplantar keratoderma, were investigated by immunocytochemistry, co-immunoprecipitation, and functional assays [133,134]. All mutants co-localized and co-immunoprecipitated with wild-type Cx26, indicating that they interact physically, likely by forming admixed heteromeric/heterotypic channels. All nine mutants inhibited the transfer of calcein in cells stably expressing Cx26, demonstrating that they each have dominant effects on wild-type Cx26. When expressed alone, all mutants formed gap junction plaques, but with impaired intercellular dye transfer. When expressed with Cx30, all mutants co-localized and co-immunoprecipitated with Cx30, indicating they likely co-assembled into heteromers. Furthermore, most of the Cx26 mutants inhibited the transfer of neurobiotin or calcein, indicating that these Cx26 mutants have trans-dominant effects on Cx30 [133,134].

Rodriguez-Paris et al. (2016) compared two different variants at the same amino acid position Gly45: a novel non-syndromic variant p.Gly45Arg, found in an individual with a likely dominantly inherited post-lingual mild hearing loss, and p.Gly45Glu, implicated in the rare fatal KID syndrome. Variant p.Gly45Arg resulted in the formation of dysfunctional gap junctions that selectively affect the permeation of negatively charged IP3 (inositol 1,4,5-trisphosphate), while p.Gly45Glu resulted in abnormal protein trafficking to the cell membrane, dysfunctional hemichannels, and the lack of gap junctions in the presence of wild-type protein [135].

Choi et al. (2009) studied the localization and gap junction functions of a novel dominant variant p.Asp46Glu, found in Korean families. The Cx26-Asp46Glu mutant was targeted to the plasma membrane, but it failed to transfer Ca^2+^ or PI (propidium iodide) intercellularly, suggesting disruption of both ionic and biochemical coupling. Heterozygous channels also showed dysfunctional intercellular couplings and hemichannel opening, confirming the dominant-negative nature of the p.Asp46Glu variant [136].

**Deafness with palmoplantar keratoderma (PPK).** In the EL1 domain, there are three dominant variants associated with deafness accompanied by PPK, for which functional studies have been performed: two are located at position Gly59 (p.Gly59Ala and p.Gly59Val) and one at position His73 (p.His73Arg).

The p.Gly59Ala mutant, expressed in communication-deficient HeLa cells, exerted impaired intracellular trafficking and targeting to the plasma membrane, and the p.Gly59Ala mutant protein was incorporated into gap junction plaques when co-expressed with either wtCx26 or wtCx30, suggesting that the proteins can oligomerize to form connexons. However, the p.Gly59Ala appears to impede dye transfer when present in gap junction plaques with Cx30 [132]. Forge et al. (2003) found that p.Gly59Ala had a perinuclear localization when expressed alone but was trafficked to the membrane when co-expressed with Cx30. Co-expression of p.Gly59Ala with Cx30 significantly reduced neurobiotin transfer in comparison with cells expressing Cx30 only. These results indicate that Cx26 and Cx30 can oligomerize to form heteromeric connexons and demonstrate a dominant-negative effect of some Cx26 mutants on Cx30 [137]. The mutant Gly59Ala protein was transported to the cell surface but did not form gap junctions that are permeable to small fluorescent dyes. When co-expressed with wild-type Cx26, the mutant Gly59Ala exerted dominant-negative effects on Cx26 function and a trans-dominant-negative effect on co-expressed wild-type Cx32 and Cx43 [138]. The properties of p.Gly59Ala, along with other Cx26 PLP variants, were also investigated in the comparative studies by Yum et al. and Zhang et al. [133,134].

Another variant at position Gly59, p.Gly59Val, was investigated together with other Cx26 mutations, in the *Xenopus* oocytes expression system, in the study by Palmada et al. (2006) [118]. Cx26 hemichannel activity was measured by depolarization-activated conductance in non-coupled oocytes. All mutants, including p.Gly59Val, showed a partially or completely defective phenotype. Co-expression of wild-type and mutant Cx26 injected at equimolar levels revealed that p.Gly59Val did not exert a dominant inhibitory effect, whereas when co-expressed with Cx30, a connexin partially co-localized with Cx26 in the cochlea, all mutants had a dominant behavior [118].

The studies of variant p.His73Arg, located at position His73, revealed that the His73Arg mutant protein failed to form gap junction channels or hemichannels when expressed alone. Co-expression of this mutant with wild-type Cx43 showed a trans-dominant inhibition of Cx43 gap junction channels, without reductions in Cx43 protein synthesis. The formation of heteromeric connexons resulted in significantly increased Cx43 hemichannel activity in the presence of the His73Arg mutant protein. These findings suggest a common mechanism whereby Cx26 mutations causing PPK and deafness trans-dominantly influence multiple functions of wild-type Cx43 and implicate a role for aberrant hemichannel activity in the pathogenesis of PPK, and further highlight an emerging role for Cx43 in genetic skin diseases [139,140].

**KID syndrome.** To date, four different dominant variants in the EL1 domain associated with KID syndrome have been functionally analyzed. Three of them are different amino acid substitutions at position Asp50 (p.Asp50Asn, p.Asp50Tyr, and p.Asp50Ala) and one at position Gly45 (p.Gly45Glu).

The p.Asp50Asn is the most frequent pathogenic *GJB2* variant associated with KID syndrome. In addition, van Geel et al. (2002) reported that p.Asp50Asn was also associated with another syndrome, called hystrix-like ichthyosis deafness (HID) syndrome, which strongly resembles KID syndrome. These disorders are distinguished mainly on the basis of electron microscopic findings [141]. The effects of p.Asp50Asn, as well as several other Cx26 syndromic dominant variants, on channel function were explored in the study by Lee et al. [94]. The proteins were all expressed in *Xenopus* oocytes with levels equal to wild-type Cx26. The p.Asp50Asn variant resulted in larger hemichannel currents than the wild-type-expressing cells. Elevated hemichannel activity correlated with an increased cell death. This result could be reversed through the elevation of calcium in the extracellular media. This set of data confirmed that aberrant hemichannel activity is a common feature of KID-associated Cx26 variants, and this may contribute to a loss of cell viability and tissue integrity [94]. In the study by Terrinoni et al. (2010), it was demonstrated that two different Cx26 mutants (Cx26-Asp50Asn and Cx26-Gly11Glu) cause cell death in vitro by the alteration of intracellular calcium concentrations. These results help to explain the pathogenesis of both the hearing and skin phenotypes, since calcium is also a potent regulator of the epidermal differentiation process. [92]. Sánchez et al. (2013) found that variant p.Asp50Asn produced multiple aberrant hemichannel properties, including a loss of inhibition by extracellular Ca^2+^, decreased unitary conductance, increased open hemichannel current rectification, and voltage-shifted activation. They demonstrated that Asp50 is a pore-lining residue and that negative charge at this position strongly influences open hemichannel properties. These data indicate that Asp50 exerts effects on Cx26 hemichannel and channel function as a result of its dual role as a pore residue and a component of an inter-subunit complex in the extracellular region of the hemichannel. Differences in the effects of substitutions in gap junction channels and hemichannels suggest that perturbations in structure occur upon hemichannel docking that significantly impact function [142]. Press et al. (2017) investigated p.Asp50Asn along with other dominant Cx26 mutants linked to various syndromic or non-syndromic diseases and demonstrated that when gap-junction-deficient HeLa cells expressed the p.Asp50Asn mutant, they underwent cell death. These findings suggest that Cx26 mutants that promote cell death or exert trans-dominant effects on other connexins in keratinocytes will lead to skin diseases and hearing loss, whereas mutants having reduced channel function but exhibiting no aberrant effects on co-expressed connexins cause only hearing loss. Moreover, cell-death-inducing *GJB2* mutations lead to more severe syndromic disease [100].

Another variant, p.Asp50Tyr, at position Asp50, associated with KID syndrome, resulted in the formation of aberrant hemichannels that might result in elevated intracellular calcium levels, a process that may contribute to the hyperproliferative epidermal phenotypes of KID syndrome [106]. Lopez et al. (2013) investigated the mechanism by which extracellular Ca^2+^ regulates the opening and closing of unpaired Cx26 hemichannels in the plasma membrane and found that aspartate to asparagine or tyrosine substitutions at position D50 (Asp50Asn/Tyr) severely compromise the ability of Cx26 hemichannels to be regulated by extracellular Ca^2+^. Analysis of the kinetic and steady-state data strongly suggests that D50 stabilizes the open state of Cx26 hemichannels by interacting with a positively charged residue (K61) in the adjacent connexin subunit. Disruption of this interaction by extracellular Ca^2+^ can facilitate destabilization of the open state and promotes hemichannel closing [143].

The third variant, p.Asp50Ala, at position Asp50, associated with KID syndrome, is less common. Mhaske et al. (2013), using three different expression systems (cRNA-injected *Xenopus* oocytes, transfected HeLa cells, or transfected primary human keratinocytes), examined the functional characteristics of two pathogenic variants causing either mild (p.Asp50Ala) or lethal (p.Ala88Val) KID syndrome. They showed that both Cx26-Asp50Ala and Cx26-Ala88Val form active hemichannels that significantly increase membrane current flow compared with wild-type Cx26. This increased membrane current accelerated cell death in low extracellular calcium solutions and was not due to increased mutant protein expression. Elevated mutant hemichannel currents could be blocked by an increased extracellular calcium concentration. These results show that these two variants exhibit a shared gain of functional activity and support the hypothesis that increased hemichannel activity is a common feature of human Cx26 mutations responsible for KID syndrome [144].

The missense variant p.Gly45Glu (c.134G > A) is associated with a particular form of KID syndrome, which is often fatal in the first year of life because of cutaneous infections and septicemia [145,146]. The effects of p.Gly45Glu have been elucidated in several functional studies [125,126,127,135,147,148,149].

Stong et al. (2006) investigated the properties of variant p.Gly45Glu expressed in HEK293 cells. The primary effect of p.Gly45Glu was causing leaky hemichannels (abnormally open hemichannels at a normal Ca^2+^ concentration) that resulted in apoptosis and cell death within 24 h of transfection. Increasing the concentration of extracellular calcium rescued the cells in a dose-dependent manner. The rescued cells formed functional gap junction channels permeable to both ions and fluorescent tracer molecules. These data showed that abnormally open hemichannels with resultant cell death, in addition to channel and hemichannel uncoupling, is a novel molecular mechanism by which Cx26 mutations may result in hearing impairment [127]. Gerido et al. (2007), using the *Xenopus* oocyte expression system to examine the functional characteristics of p.Gly45Glu, showed that oocytes were able to express both wild-type Cx26 and p.Gly45Glu variants, each of which formed hemichannels and gap junction channels. However, Cx26-Gly45Glu hemichannels displayed significantly greater whole-cell currents than wild-type Cx26, leading to cell lysis and death. This severe phenotype could be rescued in the presence of elevated extracellular Ca^2+^. The Cx26-Gly45Glu could also form intercellular channels with a similar efficiency as wild-type Cx26; however, with increased voltage-sensitive gating [125]. Sanchez et al. (2010) quantified the Ca^2+^ sensitivity and examined the biophysical properties of the p.Gly45Glu mutant at macroscopic and single-channel levels and found that the p.Gly45Glu hemichannels exhibited a substantial increase in permeability to Ca^2+^ [126]. Mese et al. (2011) created an animal model for KID syndrome by generating an inducible transgenic mouse expressing Cx26-Gly45Glu in keratinocytes. The abnormalities in Cx26-Gly45Glu mice (reduced viability, hyperkeratosis, scaling, skin folds, and hair loss) correlated with human KIDs pathology and were associated with increased hemichannel currents in transgenic keratinocytes. These results confirm the pathogenic nature of p.Gly45Glu and provide a new model for studying the role of aberrant connexin hemichannels [147]. Zhang et al. (2013) tested the hypothesis that glycine at position 45 is an important component of the sensor regulating the Ca^2+^ gating of gap junction hemichannels among multiple Cx subtypes expressed in the cochlea. They investigated, using an in vitro expression system (HEK 293 cells), the functional effects of p.Gly45Glu in other connexins (Cx30, Cx32, and Cx43). They found that p.Gly45Glu in Cx30 resulted in similar deleterious effects (leaky hemichannels) as the same variant in Cx26. The cell death occurred within 24 h of transfection, which was rescued by increasing the extracellular Ca^2+^. Whole-cell membrane current recordings indicated that p.Gly45Glu caused increased hemichannel activities. Variants p.Gly45Glu of Cx32 and Cx43 also resulted in leaky hemichannels compared to their respective wild-types in lower Ca^2+^ [148]. Ogawa et al. (2014) reported the first instance in the literature where the reversion of a “confining” nonsense mutation in the *GJB2* gene released the dominant pathogenic effect of a co-existing gain-of-function mutation, eliciting the lethal form of KID syndrome. The heterozygous missense variant p.Gly45Glu was revealed in the patient with a lethal form of KID syndrome from obviously healthy parents. The patient’s mother had the identical variant, p.Gly45Glu, which was confined by nonsense variant p.Tyr136X. An epidemiologic estimation demonstrates that approximately 11,000 individuals in the Japanese population may have the same lethal variant p.Gly45Glu but are nonetheless protected from the manifestation of the syndrome because they also inherit the common “confining” nonsense variant p.Tyr136X [149]. Rodriguez-Paris et al. (2016) showed that p.Gly45Glu resulted in abnormal protein trafficking to the cell membrane, dysfunctional hemichannels, and a lack of gap junctions in the presence of wild-type protein [135].

**Vohwinkel syndrome.** Two missense variants, p.Tyr65His and p.Asp66His, in the EL1 domain of Cx26, were found in association with Vohwinkel syndrome.

Rouan et al. (2001) studied the effects of four dominant variants, including p.Asp66His, on gap junctional intercellular communication. Mutant Cx26 variants alone and together with the epidermal connexins Cx26, Cx37, and Cx43 were expressed in paired *Xenopus* oocytes, and the intercellular coupling was measured by dual-voltage clamping. Homotypic expression of each connexin as well as co-expression of wtCx26/wtCx43 and wtCx26/wtCx37 yielded variable, yet robust, levels of channel activity. However, all four Cx26 mutants were functionally impaired and failed to induce intercellular coupling. When co-expressed with wtCx26, all four mutants suppressed the wtCx26 channel activity, consistent with a dominant inhibitory effect. However, only those Cx26 mutants associated with a skin phenotype also significantly inhibited the intercellular conductance of co-expressed wtCx43, indicating a direct interaction of mutant Cx26 units with wtCx43 [131]. Bakirtzis et al. (2003) reported the transgenic K10Connexin 26(D66H) mice model, which expressed mutant Cx26 (*GJB2*/connexin 26(D66H)), from a keratin 10 promoter, exclusively in the supra-basal epidermis (the cells in which Cx26 is upregulated in epidermal hyperproliferative states). From soon after birth, the mice exhibited a keratoderma similar to that in humans carrying the Cx26 p.Asp66His variant (true Vohwinkel syndrome). Transgene expression was associated with loss of Cx26 and Cx30 from epidermal keratinocyte intercellular junctions and accumulation in the cytoplasm. Light and electron microscopy showed marked thickening of the epidermal cornified layers and increased epidermal TUNEL staining, indicative of premature keratinocyte programmed cell death [150]. The properties of p.Asp66His were subsequently investigated in different expression systems by analysis of its intracellular localization, ability to form properly functional channels, and the effects of this mutant protein when it was co-expressed together with wild-type Cx26 and Cx30 proteins, which are co-localized within the inner ear [132], and on Cx43 protein, which is expressed in the skin [96,138]. The results evidenced that the mutant Cx26-Asp66His was functionally impaired and failed to induce intercellular coupling, and it exerted a trans-dominant-negative effect on Cx26, Cx30, and Cx43.

The functional effects of a novel variant, p.Tyr65His (c.193T > C), found in a patient with Vohwinkel syndrome, was investigated in the study by de Zwart-Storm et al. [151]. Mutant Cx26-Tyr65His protein aggregated around the nucleus, also demonstrating the presence of some punctate fluorescence and residual gap junction plaques. The localization of the mutant protein was almost completely normalized following transfection into HeLa Ohio cells stably expressing wtCx26. To determine if the gap junction channels formed by mutant Cx26 are functional, a parachute assay was performed, which showed reduced dye (calcein) transfer for p.Tyr65His [151].

**Bart–Pumphrey syndrome.** The substitution p.Asn54Lys at position Asn54 was found in association with Bart–Pumphrey syndrome—an autosomal dominant disorder characterized by sensorineural hearing loss, palmoplantar keratoderma, knuckle pads, and leukonychia, which shows considerable phenotypic variability. The clinical features partially overlap with Vohwinkel and KID syndromes. Press et al. (2017) investigated p.Asn54Lys, together with other dominant Cx26 mutants, expressed in HeLa cells, and demonstrated that the Cx26-Asn54Lys has trafficking defects, an impaired dye transfer ability, and exhibits dominant or trans-dominant properties on wild-type Cx26 and co-expressed Cx30 and Cx43 [100]. Beach et al. (2020) expressed several mutants, including p.Asn54Lys, in cochlear-relevant HEI-OC1 cells derived from the developing organ of Corti, and showed that the Cx26-Asn54Lys protein is a loss-of-function trafficking-defective mutant that can be rescued by the co-expression of Cx30, which is typically co-expressed with Cx26 in the organ of Corti [109].

### 4.4. Transmembrane Domain 2 (TM2)

The transmembrane domain 2 (TM2) of Cx26 (from 74 to 94 a.a.) includes 21 amino acids. Twelve amino acids can undergo multiple substitutions: Arg75 (3), Ala78 (2), Leu 79 (2), Gln80 (5), Leu81 (2), Ile82 (3), Val84 (4), 85Ser (2), 86Thr (2), Ala88 (5), Leu90 (3), and Met93 (3). In TM2, 40 different PLP missense variants are registered by the DVD. Among them, about half were defined as evident autosomal recessive variants, and four as dominant variants (one non-syndromic and three syndromic), while the type of inheritance for the other PLP variants remains uncertain due to their rarity and the scarce information available (Figure 4).

#### 4.4.1. Dominant Variants

Functional studies were performed for dominant variants located at amino acid positions Arg75 (p.Arg75Trp, p.Arg75Gln) and Ala88 (p.Ala88Val) of the TM2.

**Arg75.** Three different missense variants were found at amino acid position Arg75 in the TM2 domain, leading to different functional consequences: p.Arg75Trp, associated with non-syndromic dominant hearing loss (DFNA3), p.Arg75Gln, associated with deafness and palmoplantar keratoderma (PPK), and a rare variant p.Arg75Gly, associated with non-syndromic recessive hearing loss, which has not yet been functionally studied.

The dominant variant p.Arg75Trp, first found in an Egyptian family, has been extensively studied in numerous functional in vitro studies [107,119,131,132,133,134,152,153,154,155] and in mice models [156,157,158,159]. The deleterious dominant-negative effect of p.Arg75Trp on gap channel function was first demonstrated in the paired oocyte expression system: p.Arg75Trp was unable to induce electrical conductance between adjacent cells and almost completely suppressed the activity of the co-expressed wild-type Cx26 protein [152]. Marziano et al. (2003) studied p.Arg75Trp in communication-deficient HeLa cells and found a disruption of the intercellular coupling for the Arg75Trp mutant protein, and the dye transfer rate between cells co-expressing either Cx26 or Cx30 together with p.Arg75Trp was significantly reduced compared with the wild-type proteins alone [132]. Oshima et al. (2003) studied the role of p.Arg75Trp by expressing mutant connexin in insect Sf9 and HeLa cells. Gap junctions formed by p.Arg75Trp mutant showed a negligible activity in dye transfer experiments. The results suggest that the Arg75 is not important for proper membrane insertion but rather for the stabilization of the connexon structure. Co-expression of wtCx26 with the Arg75Trp mutant protein rescued the formation of stable connexon. This evidence strongly suggests that the Arg75 is responsible for inter-subunit interactions in the connexon, further supporting the model that the side chain of Arg75 makes a crucial contribution to the stability of the hexameric connexin assembly [119]. Subsequent studies [107,131,133,134,153] analyzed the features of p.Arg75Trp (subcellular localization, ability to form gap junctions, permeability and intercellular communication of gap junction channels, co-expression with other connexins) by different experimental assays and in several in vitro expression systems (HeLa cells, *Xenopus laevis* oocytes, a rat epidermal keratinocyte cell line) convincingly confirmed the dominant properties of this pathogenic variant.

The dominant variant p.Arg75Gln is another substitution at position Arg75. Hearing impairments observed in patients with p.Arg75Gln are characterized by variable severity, age of onset, and progression of hearing loss, and can be associated with variable skin alterations, starting from normal skin to severe palmoplantar hyperkeratosis [160,161,162]. Piazza et al. (2005) described the four generations of a family showing a profound sensorineural hearing loss, inherited in an autosomal dominant fashion, where p.Arg75Gln was not associated with PPK in any family member. Cell transfection and fluorescence imaging, dye transfer experiments, and dual-patch clamp recording showed that the mutant completely prevents the formation of functional channels, despite assembling into junctional plaques, in communication-incompetent HeLa cells. Due to the lack of PPC in this family, Piazza et al. suggested that p.Arg75Gln is not sufficient for the development of the complete syndromic phenotype, and the association of PPK with profound hearing loss may be dependent on the genetic background, requiring a functional interaction between the mutated Cx26 and other epidermally expressed connexins [161]. Yum et al. (2010) and Zhang et al. (2011) showed that variant p.Arg75Gln co-localized and co-immunoprecipitated with wild-type Cx26, indicating that they interact physically. Variant p.Arg75Gln inhibited the transfer of calcein in cells stably expressing Cx26, demonstrating its dominant effect on wild-type Cx26. It formed gap junction plaques, but with impaired intercellular dye transfer, when expressed alone, and it co-assembled into heteromers when expressed with Cx30, and inhibited the transfer of neurobiotin or calcein, indicating its trans-dominant effects on Cx30 [133,134]. Kim et al. (2015) presented a Korean patient with non-syndromic hearing loss caused by both the p.Arg75Gln and p.Val37Ile variants in the same allele, which was inherited from her father. Both p.Arg75Gln and Val37Ile were localized at the cell membrane and formed gap junctions between two adjacent cells. The results of the hemichannel study and dye loading suggest that the p.Val37Ile and p.Arg75Gln variants significantly reduce hemichannel activity, whereas cis mutations slightly improve hemichannel permeability [122].

**Ala88.** There are five amino acid substitutions at position Ala88. One of them, p.Ala88Val, is associated with a lethal form of KID syndrome, while the other uncommon variants, p.Ala88Ser, p.Ala88Gly, p.Ala88Glu, and p.Ala88Pro, are considered non-syndromic recessive variants. The functional characteristics of the dominant variant p.Ala88Val, associated with a lethal form of KID syndrome, were analyzed using three different expression systems (*Xenopus* oocytes, transfected HeLa cells, or transfected primary human keratinocytes) in the study by Mhaske et al. [144]. The Cx26-Ala88Val mutant protein formed active hemichannels that significantly increased membrane current flow, resulting in accelerated cell death in low extracellular calcium concentrations. Elevated mutant hemichannel currents could be blocked by increased extracellular calcium concentrations. These results support the hypothesis that increased hemichannel activity is a common feature of the Cx26 mutations responsible for KID syndrome [144].

#### 4.4.2. Recessive Variants

Numerous functional studies have been performed to better understand the role of pathogenic *GJB2* variants in the TM2 of Cx26 associated with non-syndromic recessive hearing loss (located at positions Arg77, Ile82, Val84, 86Thr, Ala88, and Leu90); however, the obtained results are sometimes inconsistent [116,117,118,119,121,124,136,152,163,164,165,166].

Richard et al. (1998) investigated variant p.Trp77Arg in the paired oocyte expression system and showed that the Cx26-Trp77Arg mutant protein failed to induce intercellular channel activity but did not significantly inhibit the ability of wild-type Cx26 to form functional channels when co-expressed. This study demonstrated that this homozygous missense variant results in a complete loss of connexin function and thus is likely to produce a disease phenotype. This variant in a heterozygous state, however, does not significantly interfere with the channel activity of the co-expressed wild-type protein and is consistent with a normal phenotype [152]. Martin et al. (1999) investigated the functional effects of three mutant Cx26 variants, including p.Trp77Arg, by examining key steps involved in the assembly of gap junctions. The p.Trp77Arg mutant protein was inefficiently targeted to the plasma membrane and retained in intracellular stores and showed limited oligomerization into connexon hemichannels [116]. Bruzzone et al. (2003) tested the ability of nine DFNB1 mutations, including p.Trp77Arg, to form channels in different expression systems and found that, although most mutations resulted in a complete loss of function, the p.Trp77Arg variant exhibited junctional currents that were virtually indistinguishable, in both magnitude and voltage-gating properties, from those determined for wtCx26. These data suggest that, although p.Trp77Arg retains the ability to form functional channels, other properties (such as unitary conductance, size selectivity, and open time probability) may be affected, thus resulting in a functional deficit [121].

Variant p.Ile82Met was detected in compound heterozygosity with c.35delG in two brothers with profound hearing impairment [167]. Palmada et al. (2006) functionally characterized more frequent *GJB2* mutations in *Xenopus* oocytes, including p.Ile82Met, identified in patients showing non-syndromic hearing impairment. Cx26 hemichannel activity was measured by depolarization-activated conductance in non-coupled oocytes, and all mutants showed a partially or completely defective phenotype. Co-expression of wild-type and mutant Cx26s injected at equimolar levels revealed that several Cx26 mutants, including p.Ile82Met, exerted a dominant inhibitory effect. When co-expressed with Cx30, all mutants had a dominant behavior [118].

Variant p.Val84Leu underwent functional analysis for the first time in the study by Bruzzone et al. [121]. The p.Val84Leu variant exhibited junctional currents that were virtually indistinguishable, in both magnitude and voltage-gating properties, from those determined for wtCx26. These data suggest that, although p.Val84Leu retained the ability to form functional channels, other properties (such as unitary conductance, size selectivity, and open time probability) may be affected, thus resulting in a functional deficit [121]. Wang et al. (2003) investigated, in communication-deficient N2A cells, the functional property of several mutant variants, including p.Val84Leu, associated with recessive DFNB1 deafness. The Cx26-Val84Leu mutant protein formed gap junctions with a junctional conductance similar to that of wild-type Cx26 junctional channels, and mutant Cx26-Val84Leu gap junctions permitted neurobiotin transfer between pairs of transfected N2A cells [163]. Beltramello et al. (2004) revealed that p.Val84Leu affects neither intracellular sorting nor electrical coupling, but specifically reduces permeability to IP3 (inositol 1,4,5-trisphosphate) [164]. The study by Zhang et al. (2005) revealed, via three independent methods, that mutant p.Val84Leu, as well as p.Ala88Ser and p.Val95Met, retained ionic coupling (Na^+^ and Ca^2+^) but blocked permeability to molecules with molecular weights larger than simple ions [124].

Functional studies of variant p.Thr86Arg, identified in Korean and Chinese individuals, revealed that the Cx26-Thr86Arg mutant protein was not expressed at the cell membrane and lacked the ability to form gap junctions. When Cx26-Thr86Arg was co-expressed with Cx26-WT, ionic and biochemical coupling was normal, consistent with the recessive nature of this variant [136,165,166].

Variant p.Leu90Pro, associated with mild to moderate hearing impairment, is common in many populations. Thönnissen et al. (2002) characterized p.Leu90Pro, along with other pathogenic *GJB2* variants, and revealed that p.Leu90Pro exerted strong membranous localization, though it did not show intercellular coupling, and oligomerization studies suggested a partly disturbed assembly of hemichannels in the p.Leu90Pro mutants [117]. However, in the study by d’Andrea et al. [88], the p.Leu90Pro mutant displayed both intracellular and membrane localization, but low levels of junctional plaques. Bruzzone et al. (2003) have analyzed the functional consequences of nine DFNB1 mutations, including p.Leu90Pro, and revealed that this mutant was unable to form intercellular channels in paired oocytes [121]. Palmada et al. [118] revealed that mutant Cx26-Leu90Pro protein did not significantly modify wild-type Cx26-mediated currents, thus corroborating the described phenotype for this variant.

### 4.5. Cytoplasmic Loop (CL)

There are 37 different missense PLP variants located in the cytoplasmic loop (CL) (from 95 to 135 a.a.) of Cx26. An association with non-syndromic autosomal recessive deafness (DFNB1) has been proven for a little over 40% of them, the only variant (p.Gly130Val) is dominant and syndromic (Vohwinkel syndrome), and the type of inheritance is uncertain for other variants due to their scarce documentation and rarity (Figure 4). Nine amino acids can undergo more than one substitution: Arg98 (3), His100 (4), Phe106 (2), Gly109 (2), Lys112 (2), Arg127 (2), Ile128 (2), Gly130 (3), and Trp133 (2). The studies of functional consequences of missense PLP variants in CL were carried out for a limited number of recessive variants [81,87,121,124,163].

Wang et al. (2003) investigated the functional properties of several Cx26 mutants implicated in recessive hearing loss, including p.Val95Met, expressed in a communication-deficient cell line N2A. The Cx26-Val95Met mutant protein failed to affect the ability of Cx26 to form homotypic gap junctions [163]. A further study by Zhang et al. (2005) revealed, via three independent methods, that mutant p.Val95Met retained ionic coupling (it was still permeable to Na^+^ and Ca^2+^) but blocked permeability to molecules with molecular weights larger than simple ions [124].

Kim et al. (2016) compared the effects of substitutions with different amino acids at the same positions in Cx26: at position His100 (p.His100Leu and p.His100Tyr) and at position Arg127 (p.Arg127Leu and p.Arg127His). All these mutant Cx26 proteins localized on the cell membrane and formed channels with no difference in the number of channels compared to wtCx26. However, no PI dye loading was observed in any of these Cx26 mutants in the hemichannel permeability assay. Therefore, the pathogenic effects of the analyzed variants on gap junction function are due to the lack of permeability/conductivity of their hemichannels [81].

Two different substitutions, p.Gly109Val and p.Gly109Glu, were found at position Gly109. One of them, recessive variant p.Gly109Val, has been firstly identified *in trans* with the p.(Glu47*) mutation [168]. Dalamon et al. (2016) performed electrophysiological and permeability studies that demonstrated that p.Gly109Val does not form functional channels but forms functional hemichannels with enhanced extracellular Ca^2+^ sensitivity and subtle alterations in voltage dependence and ATP/cationic dye selectivity [87]. There are no functional studies on variant p.Gly109Glu.

The variant p.Ser113Arg, associated with an autosomal recessive form of non-syndromic hearing loss, was found in several studies. In the study by Bruzzone et al. (2003), the ability of p.Ser113Arg (along with several other pathogenic *GJB2* variants) to form intercellular channels was tested in the paired *Xenopus* oocytes expression system. It was shown that p.Ser113Arg resulted in only background levels of junctional conductance and did not induce the formation of homotypic junctional channels [121].

### 4.6. Transmembrane Domain 3 (TM3)

Transmembrane domain 3 (TM3) (from 136 to 156 a.a.) contains 21 amino acids. There are 15 different PLP missense variants, the mode of inheritance of which has remained undetermined for about half of them (Figure 4). Three amino acids undergo multiple substitutions: Ser139 (2), Phe142 (4), and Arg143 (3). Functional studies were performed for PLP variants located at amino acid positions Phe142 (p.Phe142Leu) and Arg 143 (p.Arg143Gln and p.Arg143Trp) [85,96,118,133,134,163].

**Phe142.** Interestingly, three different nucleotide substitutions at position Phe142 (c.424T > C, c.426C > G, or c.426C > A) can result in the same replacement of phenylalanine with leucine (p.Phe142Leu). A heterozygous missense variant p.Phe142Leu was detected in the patients with unusual mucocutaneous findings and deafness without the classic features of Vohwinkel, KID, or hystrix-like ichthyosis deafness syndromes [169,170]. Another variant, p.Phe142Ile, produced by the c.424T > A substitution, was found in a heterozygous state in one patient without any mucocutaneous signs [171] and has not yet been subjected to functional analysis.

Albuloushi et al. (2020) analyzed the functional effects of p.Phe142Leu in HeLa cells expressing Cx26 or Cx43 and in HaCaT cells (model keratinocyte cell line). The Cx26-Phe142Leu mutant protein was restricted to perinuclear areas in HeLa26 cells. Cell viability of the p.Phe142Leu-transfected cells was greater in HeLa cells expressing Cx43 than in native Cx-free HeLa cells. Co-immunoprecipitation suggested a possible interaction between Cx26-Phe142Leu mutant protein and Cx43. Expression of Cx26-Phe142Leu resulted in microtubule collapse, rescued by interaction with Cx43. The *GJB2* mutations interacted with Cx43, suggesting that unique Cx43:Cx26 channels are central to the diverse phenotype of Cx26 skin-related channelopathies [96].

**Arg143.** Variant p.Arg143Gln at position Arg143 has been reported in the heterozygous state in multiple individuals, and in several families segregating with high-frequency hearing loss, indicating its dominant effect in hearing loss. The interactions of nine dominant Cx26 mutants, including p.Arg143Gln, were investigated by immunocytochemistry, co-immunoprecipitation, and functional assays [133,134]. All mutants co-localized and co-immunoprecipitated with wild-type Cx26, indicating that they interact physically, likely by forming admixed heteromeric/heterotypic channels. All nine mutants inhibited the transfer of calcein in cells stably expressing Cx26, demonstrating that they each have dominant effects on wild-type Cx26. When expressed alone, all mutants formed gap junction plaques, but with impaired intercellular dye transfer. All nine dominant Cx26 mutants were co-localized with Cx30, and most of the Cx26 mutants altered the permeability of cells expressing Cx30 to different degrees. The p.Arg143Gln diminished the transfer of calcein, indicating that this Cx26 mutant has a trans-dominant effect on Cx30 [133,134].

The p.Arg143Trp is another variant at position Arg143 that is associated with recessive non-syndromic hearing loss and found with a high frequency in the African population, suggesting a possibility of heterozygous advantage. Data generated from skin biopsies have shown that heterozygotes for p.Arg143Trp have a thicker epidermis than wild-type *GJB2* homozygotes [172]. Enhanced cellular viability of HeLa cells overexpressing p.Arg143Trp compared to those overexpressing wtCx26 has also been shown [173]. The in vitro study by Man et al. (2007) suggested an advantageous effect of p.Arg143Trp in epithelial cells. They showed that the Cx26-Arg143Trp-expressing keratinocytes formed a significantly thicker epidermis in an organotypic co-culture skin model, had increased migration compared to the wtCx26 cells, and were significantly less susceptible to cellular invasion by the enteric pathogen Shigella flexneri than the wtCx26 cells [174]. Meşe et al. (2004) analyzed variant p.Arg143Trp, along with other *GJB2* variants associated with non-syndromic recessive hearing impairment, in the paired *Xenopus* oocyte expression system. Coupling of cells expressing wild-type or mutant Cx26s was measured in the paired *Xenopus* oocyte assay. Mutant p.Arg143Trp protein was unable to form functional channels [85]. Palmada et al. (2006) functionally characterized p.Arg143Trp in *Xenopus* oocytes, together with other *GJB2* variants identified in patients showing non-syndromic hearing impairment. Cx26 hemichannel activity was measured by depolarization-activated conductance in non-coupled oocytes and all mutants showed a partially or completely defective phenotype. Co-expression of wild-type and mutant Cx26s revealed that the Cx26-Arg143Trp mutant did not exert a dominant inhibitory effect on wild-type channels. When co-expressed with Cx30, all mutants had a dominant behavior [118]. Wang et al. (2003) investigated the functional properties of several Cx26 mutants implicated in recessive hearing loss, including variant p.Arg143Trp, expressed in a communication-deficient cell line N2A. The Cx26-Arg143Trp mutant protein formed gap junctions with a junctional conductance similar to that of wild-type Cx26 junctional channels. The p.Arg143Trp gap junctions also permitted neurobiotin transfer between pairs of transfected N2A cells [163].

### 4.7. Extracellular Loop 2 (EL2)

In the second extracellular loop (EL2) domain, which includes 33 amino acids (from 157 to 189 a.a.), there are 36 missense PLP variants. About 40% of them are non-syndromic variants with evident autosomal recessive inheritance, and six variants have a dominant mode of inheritance, including five non-syndromic variants (p.Met163Leu, p.Met163Val, p.Pro175His, p.Asp179Asn, and p.Arg184Gln) and one syndromic variant, associated with deafness and palmoplantar keratoderma (PPK) (p.Ser183Phe). For other poorly documented and rare variants, the type of inheritance remains unclear (Figure 4).

#### 4.7.1. Dominant Variants

The properties of almost all (except p.Pro175His) dominant PLP variants in EL2 were investigated in several functional studies [100,109,133,134,140,175,176,177].

**Met163.** Matos et al. (2008) reported a dominant variant p.Met163Leu, first identified in a heterozygous state in a Portuguese family affected with bilateral mild/moderate high-frequency non-syndromic hearing loss. In vitro functional studies in HeLa and NEB1 cell lines demonstrated that the mutant protein harboring p.Met163Leu had defective trafficking to the plasma membrane with accumulation in the cytoplasm. In addition, the expression of Cx26-Met163Leu in HEK-293 and NEB1 cells caused an increased cell death when compared to that seen upon expression of wtCx26, and this phenotype was not rescued by a high extracellular calcium concentration. Results of the co-expression studies with wtCx26 or wtCx30 suggest that the p.Met163Leu mutant might exert a partially dominant-negative effect on wtCx26 and wtCx30 regarding cell survival [175].

Another dominant variant at position Met163, p.Met163Val, which is associated with non-syndromic (isolated) hearing loss, was investigated by Press et al. (2017), along with syndromic or non-syndromic dominant variants located in different Cx26 domains (p.Asn14Lys, p.Asp50Asn, p.Asn54Lys, and p.Ser183Phe). The p.Met163Val mutant, which causes only hearing loss, exhibited impaired gap junction function and showed no trans-dominant interactions [100].

**Asp179.** Primignani et al. (2003) described a family from southern Italy in whom autosomal dominant non-syndromic post-lingual hearing loss was associated with a heterozygous variant, p.Asp179Asn [178]. The properties of p.Asp179Asn, along with other dominant PLP variants, were also investigated in the comparative studies by Yum et al. and Zhang et al. [133,134]. The Cx26-Asp179Asn protein was able to form gap junction plaques when it was expressed alone, and it co-localized and co-immunoprecipitated with wild-type Cx26 but inhibited the transfer of calcein in cells stably expressing Cx26, demonstrating that it has dominant effects on wild-type Cx26. The Cx26-Asp179Asn protein co-localized and co-immunoprecipitated with Cx30, indicating it likely co-assembled into heteromers, and it did not diminish the transfer of neurobiotin or calcein in cells expressing Cx30 [133,134].

**Ser183.** De Zwart-Storm et al. (2008) studied variant p.Ser183Phe associated with deafness and PPK, and showed that, despite the minor defects in trafficking of part of the mutant protein to the plasma membrane, the membrane-localized channels formed by the Cx26-Ser183Phe mutant at least partially save their function, allowing transfer of calcein dye in the HeLa Ohio cell line, expressing wild-type Cx26 [176]. However, Shuja et al. (2016) showed that p.Ser183Phe was unable to form functional homomeric hemichannels or gap junctions when expressed alone. However, upon introduction to cells expressing Cx43, it reduced Cx43 gap junctional activity, altered channel gating, and increased hemichannel activity. Co-immunoprecipitation of Cx43 and mutant Cx26-Ser183Phe showed the formation of heteromeric connexons [140]. In the study by Press et al. (2017), it was shown that the p.Ser183Phe mutant formed some gap junction plaques but was largely retained within the cell and exhibited only a mild trans-dominant reduction in gap junction communication when co-expressed with Cx30. The findings of this study suggest that Cx26 mutants that promote cell death or exert trans-dominant effects on other connexins in keratinocytes will lead to skin diseases and hearing loss, whereas mutants with reduced channel function but exhibiting no aberrant effects on co-expressed connexins only cause hearing loss. Moreover, cell-death-inducing *GJB2* mutations lead to more severe syndromic disease [100]. Later, Beach et al. (2020) expressed several mutants, including p.Ser183Phe, in cochlear-relevant HEI-OC1 cells, and showed that the Cx26-Ser183Phe protein formed gap junctions incapable of transferring dye and co-localized in the same gap junctions as wild-type Cx26 and Cx30, but also gained the capacity to intermix with Cx43 within gap junctions [109].

**Arg184.** One from four known substitutions at position Arg184, p.Arg184Gln, results in non-syndromic dominant deafness, while the other three variants (p.Arg184Pro, p.Arg184Trp, and p.Arg184Gly) appear to be associated with recessive hearing loss. The heterozygous variant p.Arg184Gln was found in a patient with hearing loss inherited in a dominant manner [179]. Su et al. (2010) investigated p.Arg184Gln in Tet-On HeLa cells and demonstrated the accumulation of mutant Cx26-Arg184Gln protein in the Golgi apparatus instead of targeting to the cytoplasmic membrane. In addition, a dominant-negative effect of p.Arg184Gln on the function of wild-type Cx26 and Cx30 was shown [177]. The dominant-negative effect of p.Arg184Gln was also demonstrated in the studies by Yum et al. [133] and Zhang et al. [134].

#### 4.7.2. Recessive Variants

Seven PLP variants in EL2 (p.Phe161Ser, p.Cys169Tyr, p.Trp172Cys, p.Trp172Arg, p.Pro173Arg, p.Arg184Pro, and p.Lys188Arg) that appear to be associated with recessive hearing loss have been analyzed in numerous studies [107,109,117,121,180,181,182].

In the study by Thönnissen et al. (2002), the analysis of indirect immunofluorescence showed very weak membranous localization for mutant p.Phe161Ser, and tracer coupling experiments revealed that mutant Phe161Ser protein did not show intercellular coupling [117].

A homozygous variant, p.Cys169Tyr, was reported in association with recessively inherited profound hearing loss in an extended consanguineous Arabian family from the Middle East [183]. Zonta et al. (2015) have analyzed the effect of p.Cys169Tyr using a combination of an in vitro functional study and molecular dynamics simulations. At the cellular level, they showed that the mutant protein failed to form junctional channels in HeLa transfectants despite being correctly targeted to the plasma membrane. At the molecular level, this effect can be accounted for by disruption of the disulfide bridge that Cys169 forms with Cys64 in the wild-type Cx26 structure. The lack of the disulfide bridge in the Cx26-Cys169Tyr mutant protein causes a spatial rearrangement of two important residues, Asn176 and Thr177, which play a crucial role in the intramolecular interactions that permit the formation of an intercellular channel by the head-to-head docking of two opposing hemichannels resident in the plasma membrane of adjacent cells [180].

Two different amino acid substitutions, p.Trp172Arg and p.Trp172Cys, both associated with recessive hearing loss, were found at position pTrp172. Variant p.Trp172Arg, found in patients from India, was analyzed in the study by Mani et al. (2009). No abnormalities in the transport of mutant Cx26-Trp172Arg protein to the cell membrane were detected. Nevertheless, no LY dye transfer via gap junction channels was observed. Mani et al. suggested that the amino acid replacement that occurred near the Cys residues of EL2 may lead to failure in docking with the connexon from adjacent cells [107]. Another variant at Trp172, p.Trp172Cys, was found with a high frequency in patients from the Tyva Republic (Russia) [184]. The functional consequences of p.Trp172Cys were investigated in the study by Maslova et al. [181], where a *GJB2*-knockout HeLa cell line was established following the generation of a panel of transgenic HeLa cell lines stably expressing variant p.Trp172Cys, as well as other Cx26 mutants and wild-type Cx26. Partially disrupted trafficking of mutant Cx26-Trp172Cys protein to the cell membrane with its accumulation in cytoplasmic compartments was revealed, although a small fraction of the cells with a protein signal were observed on the cell membrane. In addition, the dye (PI)-loading assay demonstrated increased hemichannel permeability in the cell line expressing variant p.Trp172Cys compared to wild-type hemichannels [181].

Of the two variants, p.Pro173Arg and p.Pro173Ser, which were found at position Pro173, functional studies were carried out only for p.Pro173Arg [117]. Analysis of indirect immunofluorescence showed an absence of signals corresponding to p.Pro173Arg on the membrane, and the tracer-coupling experiments (by using neurobiotin injections) revealed that the Cx26-Pro173Arg mutant did not show intercellular coupling [117].

Of the three variants at position Arg184, associated with recessive deafness (p.Arg184Pro, p.Arg184Trp, and p.Arg184Gly), the functional studies were carried out only for p.Arg184Pro. Mutant p.Arg184Pro protein demonstrated impaired trafficking and accumulation in the cytoplasm, and an absence of intercellular coupling in HeLa and HEI-OC1 cells expressing p.Arg184Pro [107,109,117]. Experiments in the *Xenopus* oocytes system carried out by Bruzzone et al. revealed that injections of mutant p.Arg184Pro-Cx26 RNA into the cells did not induce the formation of homotypic channels since the levels of conductance never exceeded the background level [121].

Ambrosi et al. (2013) analyzed p.Lys188Arg along with other point mutations that cause non-syndromic hearing loss and revealed that this variant results in mis-trafficking, since an intracellular signal, corresponding to the endoplasmic reticulum or cytoplasmic accumulation of mutant protein in HeLa cells expressing the p.Lys188Arg variant, was observed [182].

### 4.8. Transmembrane Domain 4 (TM4)

The transmembrane domain 4 (TM4) (from 190 to 210 a.a.) contains 21 amino acids, and 6 of them can undergo multiple substitutions: Met195 (3), Ile196 (2), Gly200 (2), Cys202 (2), Leu205 (2), and Asn206 (3). In TM4, there are 22 different PLP missense variants registered by the DVD. The recessive mode of inheritance is well defined for half of them, and two variants are non-syndromic dominant (Figure 4). The functional consequences of PLP variants located in TM4 were investigated in several studies [81,85,86,133,134,182].

Meşe et al. (2004) analyzed recessive variant p.Asn206Ser, along with other Cx26 mutations associated with non-syndromic recessive hearing impairment, in the paired *Xenopus* oocyte expression system. The p.Asn206Ser mutant did electrically couple cells, though its voltage-gating properties were different from wild-type Cx26 channels [85]. Further permeability examination revealed that the Cx26-Asn206Ser mutant channels had impaired permeability to the cationic dye EtBr, while they retained their ability to transfer anionic LY and cAMP between cell pairs [86].

Kim et al. (2016) predicted potential structural abnormalities of several *GJB2* variants using computational modeling and performed in vitro functional analysis for confirmation. Variants Phe191Leu and p.Met195Val, located in TM4, form a hydrophobic intra/intermolecular core that contributes to structural stability, and it was predicted that these variants would affect the proper folding of Cx26, resulting in its accumulation in the cytosol. In vitro studies confirmed that the mutant Phe191Leu and Met195Val proteins were not transported to the cell membrane and stayed in the cytoplasm, particularly in the ER [81]. The properties of p.Cys202Phe, the only functionally studied dominant variant in the TM4 domain, along with other dominant *GJB2* variants, were investigated in studies by Yum et al. and Zhang et al. [133,134].

Ambrosi et al. (2013) analyzed 14 point mutations in the fourth transmembrane helix of Cx26 that cause non-syndromic hearing loss. Eight mutations caused mis-trafficking (p.Lys188Arg, p.Phe191Leu, p.Val198Met, p.Ser199Phe, p.Gly200Arg, p.Ile203Lys, p.Leu205Pro, and p.Thr208Pro). Of the remaining six that formed gap junctions in mammalian cells, p.Met195Ile and p.Ala197Ser formed stable hemichannels after isolation with a baculovirus/Sf9 protein purification system, while p.Cys202Phe, p.Ile203Thr, p.Leu205Val, and p.Asn206Ser formed hemichannels with varying degrees of instability. The function of all six gap-junction-forming mutants was further assessed through measurement of dye coupling in mammalian cells and junctional conductance in paired *Xenopus* oocytes. Dye coupling between cell pairs was reduced by varying degrees for all six mutants. Intra-hemichannel interactions between mutant and wild-type proteins were assessed in rescue experiments using baculovirus expression in Sf9 insect cells. Of the four unstable mutations (p.Cys202Phe, p.Ile203Thr, p.Leu205Val, and p.Asn206Ser), only p.Cys202Phe and p.Asn206Ser formed stable hemichannels when co-expressed with wild-type Cx26. Thus, mutations in TM4 cause a range of phenotypes of dysfunctional gap junction channels that are discussed within the context of the X-ray crystallographic structure [182].

### 4.9. C-Terminus (CT)

The C-terminus of the Cx26 protein (211–226 a.a.) contains 16 amino acids. There are eight different PLP missense variants leading to amino acid replacements in this Cx26 domain (Figure 4). The PLP variants in the C-terminus are in most cases poorly documented due to their rarity, with the exception of the p.Leu214Pro variant associated with autosomal recessive deafness, for which a functional study was performed [85]. In this study, performed in the paired *Xenopus* oocyte expression system, it was shown that the p.Leu214Pro mutant Cx26 completely lost its ability to form functional gap junction channels, although biochemical analysis showed that this mutant was expressed at similar levels to wild-type Cx26 [85].

## 5. Conclusions

Pathogenic missense variants in the *GJB2* gene, resulting in amino acid substitutions, lead to a variety of clinical outcomes, including the most common non-syndromic autosomal recessive deafness (DFNB1A), autosomal dominant deafness (DFNA3A), as well as syndromic forms combining hearing loss and skin disorders. Here, we reviewed the in vitro studies that were performed for elucidating the functional effects of different Cx26 mutants. Their biosynthesis, oligomerization into connexons, trafficking to the plasma membrane, functions of mutant hemichannels and gap junction channels, and interactions with other co-expressed connexins have been analyzed using various experimental assays.

In addition, we summarized all available data on the types of inheritance of missense PLP variants that can be classified as the evident autosomal recessive or dominant variants, as well as variants with an uncertain mode of inheritance. Dominant variants include non-syndromic (isolated hearing impairment) and syndromic (hearing impairment accompanied by mild to severe skin disorders) variants. Syndromic variants have been studied most intensively, which is likely due to them being the most severe pathological phenotypes that have attracted the interest of researchers. In general, quite a lot of relatively common recessively inherited variants have also been studied. However, the mode of inheritance and pathogenic effects remain uncertain for rare and poorly documented PLP variants, the pathogenicity of which was only predicted by in silico computational analyses but not functionally confirmed [77,78,79,80].

The distribution of PLP variants across Cx26 protein domains is uneven, which is apparently determined by the structural features and specific roles of certain Cx26 domains in intercellular gap junctions [8,9,35,38,42,62]. In addition, the varying number of amino acid positions at which multiple amino acid substitutions are observed also appears to contribute to the specific distribution of PLP variants across the Cx26 protein sequence. An assessment of the load of PLP variants (defined by normalization of their number in each Cx26 domain by the number of amino acids in the corresponding domain) showed that four Cx26 domains (N-terminus, TM1, EL1, and TM2) are most enriched in them. The N-terminus and EL1 are also enriched with syndromic dominant variants, which, as shown by corresponding functional studies, are mostly associated with ‘leaky’ hemichannels or their trans-dominant effects to co-expressed connexins; that is, they are gain-of-function variants. The recessive *GJB2* variants in homozygous or compound heterozygous states in most cases result in isolated congenital or prelingual severe to profound hearing loss. They scattered across all Cx26 domains, and as shown in many studies, the corresponding mutant Cx26 proteins generally manifest impaired folding, oligomerization, and trafficking to the plasma membrane and, finally, they are unable to form functional gap junction channels (loss-of-function variants).

In summary, this review provides updated information on the distribution of different pathogenic missense *GJB2* variants across the Cx26 domains, their patterns of inheritance, and their functional consequences, which will be useful for further research on one of the most common forms of hereditary hearing loss.

## Figures and Tables

**Figure 1 biomolecules-13-01521-f001:**
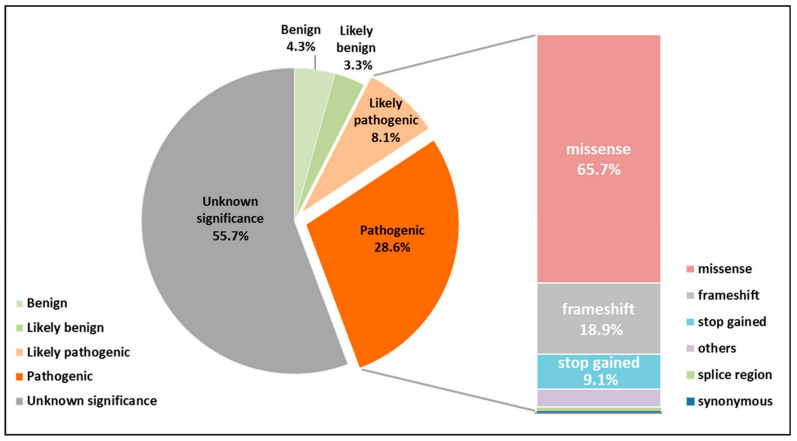
The proportions of different *GJB2* variants and the distribution of PLP variants according to their molecular consequences.

**Figure 2 biomolecules-13-01521-f002:**
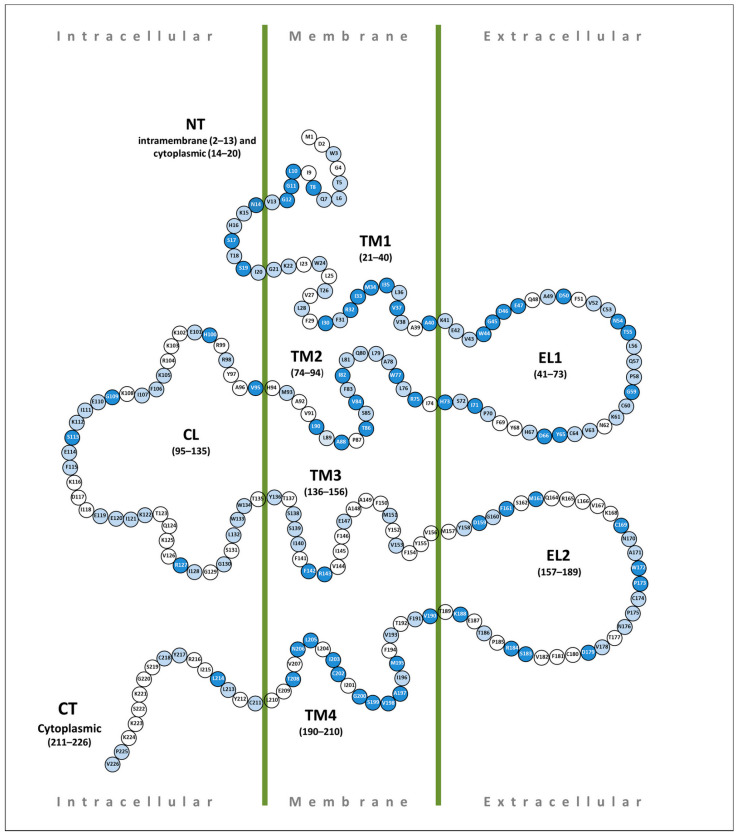
The schematic topology of the Cx26 protein protomer. Amino acids affected by known PLP missense variants leading to non-synonymous substitutions are colored (dark blue—analyzed by functional studies, light blue—without functional studies). NT—N-terminus, including intramembrane (2–13 a.a.) and cytoplasmic (14–20 a.a.) parts; TM1 (21–40 a.a.), TM2 (74–94 a.a.), TM3 (136–156 a.a.), and TM4 (190–210 a.a.)—transmembrane domains; CL (95–135 a.a.)—cytoplasmic loop; EL1 (41–73 a.a.) and EL2 (157–189 a.a.)—extracellular loops; CT—C-terminus (cytoplasmic, 211–226 a.a.). Amino acids (a.a.) positions for each domain are indicated in brackets.

**Figure 3 biomolecules-13-01521-f003:**
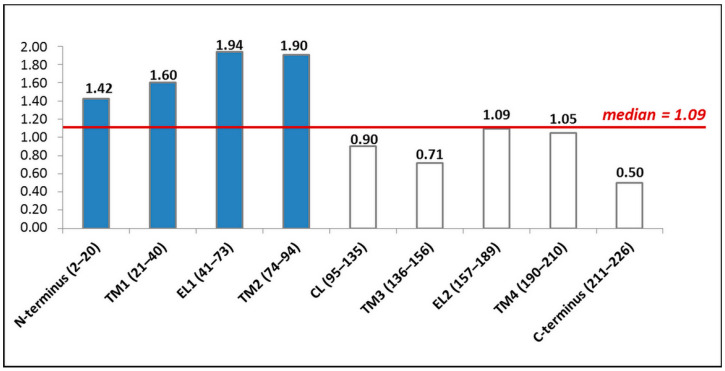
The rate of amino acid substitutions across Cx26 domains. Four Cx26 domains (N-terminus, TM1, EL1, and TM2) with a rate above the mean (median value = 1.09) are colored.

**Figure 4 biomolecules-13-01521-f004:**
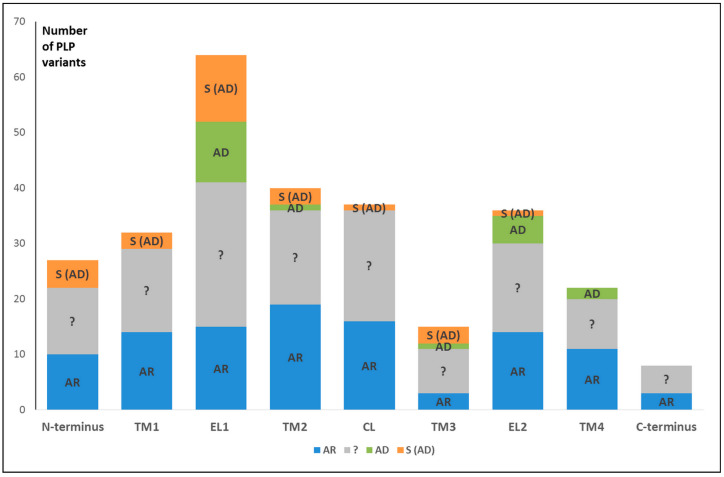
Distribution of the PLP missense variants across Cx26 domains according to their mode of inheritance. AR—non-syndromic autosomal recessive, AD—non-syndromic dominant, S (AD)—syndromic dominant, ?—uncertain type of inheritance.

## Data Availability

The data presented in this study are available in this article.

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
