# Peer review of "Functional Consequences of Pathogenic Variants of the GJB2 Gene (Cx26) Localized in Different Cx26 Domains"

_biomolecules, 2023, doi:10.3390/biom13101521_

Round 1

Reviewer 1 Report

This is an excellent and exhaustive review about the functional consequences of GJB2 pathogenic variants localized in different Cx26 domains. This article is very well documented and contains a large number of references. I have only a few comments.

Major comments

Section 2: Structure, life cycle and functions of Cx26

- Besides mutations that affect the channel function itself, many of the disease-causing mutations in GJB2 impair the trafficking and assembly of Cx26, what prevents the formation of gap junctions (Hoang Dinh et al., 2009; Xu and Nicholson, 2013). Recent findings have been made towards understanding the biogenesis of gap junction channels in the cochlea. These could represent a step toward unraveling the pathogenic significance of many of these mutations. These advances have been recently reviewed by Defourny and Thiry (Recent insights into gap junction biogenesis in the cochlea, Developmental Dynamics, 2022). In the cochlea, Cx26 and Cx30 co-assemble into two types of gap junctions, which form a syncytium extending from the spiral limbus to the cochlear spiral ligament. On the one hand, Cx30 mainly forms homomeric channels between adjacent Deiters' cells, that is, the supporting cells which surround the outer sensory hair cells. On the other hand, Cx30 co-assembles with Cx26 into heterotypic and heteromeric channels which connect the other non-sensory supporting cell types. It has been shown that gap junction biogenesis pathway in the cochlea differs according to whether Cx30 co-assembles into homomeric channels or co-assembles with Cx26 into heterotypic and heteromeric channels (Defourny et al., Cochlear connexin 30 homomeric and heteromeric channels exhibit distinct assembly mechanisms, Mechanisms of Development, 2019). In this sense, please note that ephrin-B2 mostly regulates the biogenesis of Cx30 homomeric channels between adjacent Deiters’cells, rather than that of Cx26/Cx30 heterotypic and heteromeric channels connecting the other supporting cell types.

Minor comments

- Lines 129, 131: Maybe you could chose between “post-translation modifications” or “posttranslational modifications”. I would prefer “posttranslational modifications”.

- Lines 137,139,140: please replace “assosiated” by “associated”

- Line 788: a space is needed between “with” and “Cx26”

- Line 1136: The functional consequences of p.Trp172Cys « were » instead of « was »

The quality of English language is good.

Author Response

Reviewer 1

This is an excellent and exhaustive review about the functional consequences of GJB2 pathogenic variants localized in different Cx26 domains. This article is very well documented and contains a large number of references. I have only a few comments.

Authors: Thank you very much for your positive evaluation of our paper! We also sincerely appreciate the time and effort you have devoted to providing valuable feedback on our manuscript.

Major comments

Section 2: Structure, life cycle and functions of Cx26

- Besides mutations that affect the channel function itself, many of the disease-causing mutations in GJB2 impair the trafficking and assembly of Cx26, what prevents the formation of gap junctions (Hoang Dinh et al., 2009; Xu and Nicholson, 2013). Recent findings have been made towards understanding the biogenesis of gap junction channels in the cochlea. These could represent a step toward unraveling the pathogenic significance of many of these mutations. These advances have been recently reviewed by Defourny and Thiry (Recent insights into gap junction biogenesis in the cochlea, Developmental Dynamics, 2022). In the cochlea, Cx26 and Cx30 co-assemble into two types of gap junctions, which form a syncytium extending from the spiral limbus to the cochlear spiral ligament. On the one hand, Cx30 mainly forms homomeric channels between adjacent Deiters' cells, that is, the supporting cells which surround the outer sensory hair cells. On the other hand, Cx30 co-assembles with Cx26 into heterotypic and heteromeric channels which connect the other non-sensory supporting cell types. It has been shown that gap junction biogenesis pathway in the cochlea differs according to whether Cx30 co-assembles into homomeric channels or co-assembles with Cx26 into heterotypic and heteromeric channels (Defourny et al., Cochlear connexin 30 homomeric and heteromeric channels exhibit distinct assembly mechanisms, Mechanisms of Development, 2019). In this sense, please note that ephrin-B2 mostly regulates the biogenesis of Cx30 homomeric channels between adjacent Deiters’cells, rather than that of Cx26/Cx30 heterotypic and heteromeric channels connecting the other supporting cell types.

Authors: Thank you very much for your important comment. We have studied this certainly interesting information. However, in keeping with the main topic of our review, and also in view of the already very large volume of text, we only added this reference [70, Defourny, Thiry, 2023] and slightly modified the phrase “Unlike many other connexins, Cx26 appears to lack a PDZ-binding motif [68], so its organization into gap junction plaques may apparently be regulated by other proteins [70]”.

Minor comments

- Lines 129, 131: Maybe you could chose between “post-translation modifications” or “posttranslational modifications”. I would prefer “posttranslational modifications”.

- Lines 137,139,140: please replace “assosiated” by “associated”

- Line 788: a space is needed between “with” and “Cx26”

- Line 1136: The functional consequences of p.Trp172Cys « were » instead of « was »

Authors: All of the above have been fixed. In addition, we carefully checked the text and fixed other typos and errors.

Reviewer 2 Report

This is a comprehensive review of Cx26 mutations associated with human disease (mainly deafness). The authors identified classes of mutations that are pathogenic or likely pathogenic (PLP) that were then described in detail, with particular emphasis on functional studies. The review also provides the background material needed to interpret the Cx26 literature. I anticipate this will be a well referenced review.

My only comment is that Figure S1 should be incorporated into the main body of the text, a supplement with a single figure will tend to get lost and it provides interesting information related to Cx26 mutation frequency. 

In general, the manuscript was very well written. There are a few minor grammatical errors, but this is not a major concern.

Author Response

Reviewer 2

Comments and Suggestions for Authors

This is a comprehensive review of Cx26 mutations associated with human disease (mainly deafness). The authors identified classes of mutations that are pathogenic or likely pathogenic (PLP) that were then described in detail, with particular emphasis on functional studies. The review also provides the background material needed to interpret the Cx26 literature. I anticipate this will be a well referenced review.

Authors: Thank you very much for your positive evaluation of our paper and constructive comment that has allowed us to improve our manuscript.

My only comment is that Figure S1 should be incorporated into the main body of the text, a supplement with a single figure will tend to get lost and it provides interesting information related to Cx26 mutation frequency. 

Authors: We are grateful to you for this helpful comment. We have incorporated Figure S1 into the main body of the text as Figure 3 (new).

Comments on the Quality of English Language

In general, the manuscript was very well written. There are a few minor grammatical errors, but this is not a major concern.

Authors: We have thoroughly proofread the manuscript and tried to correct all grammatical errors and typos.
